# No Change, No Gain: Empowering Graph Neural Networks with Expected Model Change Maximization for Active Learning

**Zixing Song**
The Chinese University of Hong Kong
New Territories, Hong Kong SAR
zxsong@cse.cuhk.edu.hk

**Yifei Zhang**
The Chinese University of Hong Kong
New Territories, Hong Kong SAR
yfzhang@cse.cuhk.edu.hk

**Irwin King**
The Chinese University of Hong Kong
New Territories, Hong Kong SAR
king@cse.cuhk.edu.hk

## Abstract

Graph Neural Networks (GNNs) are crucial for machine learning applications with graph-structured data, but their success depends on sufficient labeled data. We present a novel active learning (AL) method for GNNs, extending the Expected Model Change Maximization (EMCM) principle to improve prediction performance on unlabeled data. By presenting a Bayesian interpretation for the node embeddings generated by GNNs under the semi-supervised setting, we efficiently compute the closed-form EMCM acquisition function as the selection criterion for AL without re-training. Our method establishes a direct connection with expected prediction error minimization, offering theoretical guarantees for AL performance. Experiments demonstrate our method's effectiveness compared to existing approaches, in terms of both accuracy and efficiency.

## 1 Introduction

Graph Neural Networks (GNNs) have gained significant recognition in machine learning applications, particularly for graph-structured data [71, 32, 68]. Nevertheless, their efficacy is predominantly contingent on the availability of ample labeled data. The labeling process, often demanding human intervention and domain knowledge, poses high costs in real-world applications. Active Learning (AL) presents a compelling solution to mitigate the issue of limited labeled data [1]. AL boosts the performance of passive learning by iteratively training a model based on the current set of labeled nodes and selecting the query nodes to expand this set based on the different query heuristics or the designed acquisition function. The main idea is to identify the most informative nodes for the oracle to annotate, thereby enhancing test node predictions upon their addition to the training set.

Recently, active learning methods for graph-structured data have seen an upsurge, adapting general active learning techniques to accommodate the non-IID nature of graphs [88, 89, 90]. Predominantly, these methods employ heuristic approaches, selecting nodes based on rudimentary measures such as uncertainty and influence score. While efficient, these strategies lack a direct correlation with expected prediction performance on remaining unlabeled nodes—our primary concern. Hence, We advocate for performance-based active learning methods, selecting nodes that directly optimize anticipated model performance [4].To balance efficiency, we employ the Expected Model Change Maximization

37th Conference on Neural Information Processing Systems (NeurIPS 2023).

(EMCM) algorithm [6, 7] from general active learning. EMCM chooses nodes maximizing expected model change in parameters or predictions, serving as a surrogate for expected prediction error.

Implementing the EMCM principle in GNNs is both challenging and meaningful. First, the acquisition function for EMCM calculates the expected model change following the addition of a single labeled node, without the need for re-training. Hence, a Bayesian probabilistic model is preferred in AL [2]. This approach allows for efficient updating of posterior beliefs about the model's parameters or predictions upon introducing a new labeled node, aligning perfectly with performance-based methods like EMCM. However, conventional GNNs lack Bayesian interpretation, while Bayesian GNNs impose significant computational demands, rendering precise approximation of the expected model change without re-training practically impossible. Therefore, a lightweight Bayesian probabilistic learning framework could enable the application of EMCM on graphs. Second, the direct adaptation of EMCM for GNNs, despite its computational advantages over other performance-based methods, remains impractical due to the large number of nodes, which is common in real-world cases. Third, a deep connection can be established between the EMCM method on graphs and the expected prediction error, offering a theoretical guarantee regarding the potentially optimal AL performance.

In light of these insights, we are the first to extend the EMCM principle to GNNs, contributing in the following ways. First, we revisit the training process of the Simplified Graph Convolution (SGC) model [68] for the semi-supervised node classification as an example to obtain an equivalent view of a bi-level optimization problem. This motivates us to propose a regularized single-level optimization learning framework that possesses a clear Bayesian interpretation. Second, we derive the posterior mean and variance of the node embeddings projected onto the truncated graph spectral subspace via the Laplace approximation. This enables the efficient computation of the acquisition function of EMCM for graphs through a closed-form solution. Third, we provide theoretical insights into the proposed method, essentially equating to the selection of the node that directly minimizes the expected prediction errors of the remaining unlabeled nodes, assuming its addition to the training set. This theoretical interpretation aligns with the ultimate goal of AL. Fourth, We perform comprehensive experiments across several datasets, demonstrating the efficacy and efficiency of our proposed method.

## 2 Preliminary

### 2.1 Problem Formulation

We consider a graph $\mathcal{G} = (\mathcal{V}, \mathcal{E})$ with $|\mathcal{V}| = n$ nodes. Node feature vectors are represented as $\boldsymbol{x}_i \in \mathbb{R}^d$, aggregated into a node feature matrix $\mathbf{X} = [\boldsymbol{x}_1^\intercal, \cdots, \boldsymbol{x}_n^\intercal] \in \mathbb{R}^{n \times d}$. The adjacency matrix of $\mathcal{G}$ is given as $\mathbf{A} \in \{0,1\}^{n \times n}$ ($\mathbf{A} \in \mathbb{R}^{n \times n}$ if the graph is weighted), aligned with edge set $\mathcal{E}$. Assume there exists a labeling oracle that can map each node $i$ ($i \in \mathcal{V} = \{1, \cdots, n\}$) to its ground-truth one-hot label vector $\boldsymbol{y}_i \in \{0,1\}^c$. $c$ is the number of classes. The node set $\mathcal{V}$ is partitioned into the labeled node set $\mathcal{V}_l$ ($|\mathcal{V}_l| = n_l$) and the unlabeled node set $\mathcal{V}_u$ ($|\mathcal{V}_u| = n_u$). At first, only the ground-truth labels of labeled nodes $\{\boldsymbol{y}_i\}_{i \in \mathcal{V}_l}$ are revealed to the learning model.

We investigate the pool-based active learning setting on the aforementioned graph (Appendix (D.1)). The learning model initially undergoes standard training on the labeled set $\mathcal{V}_l$, $\{\boldsymbol{y}_i\}_{i \in \mathcal{V}_l}$. Subsequently, it identifies a query node set $\mathcal{V}_q$ from $\mathcal{V}_u$ ($\mathcal{V}_q \subset \mathcal{V}_u$) and solicits the labeling oracle for associated labels. Following this, the labeled and unlabeled sets are updated ($\mathcal{V}_l \leftarrow \mathcal{V}_l \cup \mathcal{V}_q, \mathcal{V}_u \leftarrow \mathcal{V}_u \setminus \mathcal{V}_q$). This training cycle and query node selection process repeats until the labeling budget $B$ is depleted ($|\mathcal{V}_l| = n_l + B$). We distinguish between sequential active learning, where one node is selected to query the oracle ($|\mathcal{V}_q| = 1$), and batch active learning, where a batch of $b$ nodes is chosen ($|\mathcal{V}_q| = b \in \mathbb{N}^+$). The primary objective is to deduce the soft labels of nodes within the unlabeled set $\{\hat{\boldsymbol{y}}_i\}_{i \in \mathcal{V}_u}$ ($\hat{\boldsymbol{y}}_i \in [0,1]^c$), utilizing the labeling oracle under a total labeling budget $B$.

More specifically, in each query node selection step under the transductive setting, we aim to solve Eq. (1) to choose the most informative node(s) and minimize the expected prediction error on the remaining unlabeled nodes. Here, $\ell \colon \mathbb{R}^c \times \mathbb{R}^c \to \mathbb{R}^+$ signifies the selected loss function.

$$\underset{\mathcal{V}_q : |\mathcal{V}_q| = b}{\arg \min} \; \mathbb{E}_{i \in \mathcal{V}_u \setminus \mathcal{V}_q}[\ell(\hat{\boldsymbol{y}}_i, \boldsymbol{y}_i)] \tag{1}$$

## 2.2 Graph Neural Networks for Semi-supervised Node Classification

Graph Neural Networks (GNNs) excel in learning representations for graph-structured data [71, 62, 10, 57, 40, 95, 43, 92, 11, 61, 94, 59, 82, 60, 58, 39, 93, 9]. Among them, the Graph Convolutional Networks (GCN) [32] model is the most representative method. Considering the GCN or other GNN models for the node classification task under the semi-supervised setting, there are two main steps during one iteration: (1) Forward pass that fuses both node features $\mathbf{X}$ and structure information $\mathbf{A}$ into the low-dimensional representation or node embeddings $\mathbf{U}(\boldsymbol{\Theta}) \in \mathbb{R}^{n \times c}$ (before the softmax function) with the training parameters $\boldsymbol{\Theta}$; and (2) Backward pass that updates $\boldsymbol{\Theta}$ through gradient decent according to the training loss function over all labeled nodes as $\min_{\boldsymbol{\Theta}} \sum_{i \in \mathcal{V}_l} \ell(\mathsf{S}(\mathbf{U}_i(\boldsymbol{\Theta})), \boldsymbol{y}_i)$. Here, $\mathbf{U}_i(\boldsymbol{\Theta})$ denotes the $i$-th row of $\mathbf{U}(\boldsymbol{\Theta})$. Typically, $\ell(\cdot, \cdot)$ is set as the cross-entropy loss after applying the Softmax function $\mathsf{S}(\cdot)$ on node embedding $\mathbf{U}_i(\boldsymbol{\Theta})$. Different GNNs mainly have different designs in the forward pass, but the backward pass remains the same in general. For a $K$-layer GCN, we optimize the training loss such that $\mathbf{U}(\boldsymbol{\Theta}) = \hat{\tilde{\mathbf{A}}}(\phi(\hat{\tilde{\mathbf{A}}}(\cdots \phi(\hat{\tilde{\mathbf{A}}} \mathbf{X} \boldsymbol{\Theta}^{(0)}) \cdots) \boldsymbol{\Theta}^{(K-2)})) \boldsymbol{\Theta}^{(K-1)}$. Here $\tilde{\mathbf{A}} = \mathbf{A} + \mathbf{I}$ represents the adjacency matrix with the added self-loop and $\tilde{\mathbf{D}} = \mathbf{D} + \mathbf{I}$ with the diagonal degree matrix $\mathbf{D} = \text{diag}(d_1, \cdots, d_n)$ where $d_j = \sum_j \mathbf{A}_{i,j}$. The normalized adjacency matrix is $\hat{\tilde{\mathbf{A}}} = \tilde{\mathbf{D}}^{-1/2} \tilde{\mathbf{A}} \tilde{\mathbf{D}}^{-1/2}$. We denote $\boldsymbol{\Theta} = \{\boldsymbol{\Theta}^{(i)}\}_{i=0}^{K-1}$. Note that GCN conducts linear transformation and non-linearity activation $\phi(\cdot)$ repeatedly. SGC [68] reduces this excess complexity by removing non-linearities and collapsing training parameters $\boldsymbol{\Theta}$ between consecutive layers. Similarly, a $K$-layer SGC also optimizes the training loss such that the node embedding matrix $\mathbf{U}(\boldsymbol{\Theta})$ is fixed as $\mathbf{U}(\boldsymbol{\Theta}) = \hat{\tilde{\mathbf{A}}}^K \mathbf{X} \boldsymbol{\Theta}$.

## 2.3 General Active Learning and Active Learning on Graphs

Active Learning (AL) [55] selects the most "informative" samples for querying, falling into three categories: uncertainty-based, representativeness-based, and performance-based methods. Uncertainty-based methods target the most uncertain instances [83, 101, 50], using criteria like entropy and margin. Techniques like Query by Disagreement (QBD) and Query by Committee (QBC) [56, 16, 52] also reside here, employing uncertainty sampling to reduce version space. Representativeness-based methods select samples best representing the input distribution, using methods such as density-weighting [14, 46, 27, 28] and clustering [49, 13, 72]. Performance-based methods directly optimize informativeness via Eq. (1) or surrogates, considering the impact of revealing an instance's label on future outcomes. Techniques like Expected Error Reduction (EER) [53], Expected Variance Reduction (EVR) [54], and Expected Model Change Maximization (EMCM) [6] are examples.

There is a recent surging trend to explore AL strategies on graph-structured data. It is unsuitable to directly apply general AL techniques to GNNs since the nodes are not i.i.d. but linked with edges such that connected nodes tend to have the same label. AGE [4] and ANRMAB [18] incorporate node embedding density and PageRank centrality into their node selection. GEEM [51] adapts the Expected Error Reduction (EER) method for graphs, while RIM [89] accounts for noisy oracles in node labeling. Furthermore, ALG [88] optimizes the effectiveness of all nodes influenced by GNNs, and IGP [90] maximizes information gain propagation. Detailed reviews are found in Appendix B.

## 2.4 Motivations and Challenges

### 2.4.1 Bayesian Probabilistic Interpretations

Active learning strategies for graphs often directly utilize GCN or other GNN models as their backbone [4, 88, 89, 90, 37], capitalizing on their exemplary representation learning abilities. However, under semi-supervised settings, these models typically lack a Bayesian probabilistic interpretation, a fundamental aspect of AL that quantifies uncertainty for uncertainty-based methods and lays the groundwork for performance-based methods, which incorporate prior knowledge and update the posterior beliefs about the model parameters or predictions, assuming if new labeled data becomes available. Relying on GCN or GNN models without Bayesian interpretations may lead to inefficient sample selection and hinder learning progress. Therefore, integrating a Bayesian probabilistic interpretation within the GNN framework is a desirable enhancement to active learning methodologies. Existing Bayesian GNNs provide well-defined Bayesian interpretations [96, 24], yet they often come with significant computational costs. This renders them impractical for active learning environments,

where model re-training is imperative after each query set selection. To address these challenges, we aim to introduce a lightweight, general Bayesian learning framework for GNN models. This will facilitate subsequent query node set selection by enabling rigorous, informed decision-making [17].

### 2.4.2 Efficient Computation

While numerous existing graph active learning methods extend uncertainty-based or representativeness-based methods for their simplicity and efficiency (Table 6 in Appendix B.2), these approaches can be sensitive to outliers and largely reliant on prerequisite density estimation functions or clustering algorithms. We instead advocate for performance-based methods, which directly examine the potential impact on the model should an unlabeled node's label be revealed. Well-established methods such as EER [53] and EVR [54] and their graph-based adaptations [51] are notorious for their extremely high computational overhead. As an alternative, we concentrate on the EMCM concept for AL on graphs. This approach selects nodes that yield the maximum expected model change, striking a balance between theoretical interpretations and computational costs. Our proposed general Bayesian learning framework for GNN models enables efficient computation of expected model changes using closed-form solutions. For graphs with an exceptionally large number of nodes, we further employ a spectral approximation technique to curtail computational costs.

### 2.4.3 Solid Theoretical Guarantees

A significant limitation of most current AL techniques for graphs is the absence of theoretical connections with the expected prediction errors in Eq. (1), the ultimate goal in AL. CSAL [65] offers an in-depth theoretical analysis of label complexity within the context of AL on graphs, but it only verifies the existence of such an algorithm without presenting a concrete implementation. This gap between theory and practice hinders the direct applicability of CSAL. To deploy AL algorithms in real-world situations, it is crucial to devise methods rooted in rigorous theoretical foundations that address this limitation, thereby enabling more efficient learning with fewer labeled examples.

## 3 Methodology

Our starting point is to propose a Bayesian probabilistic learning framework for semi-supervised node classification task based on GNNs. Instead of following the existing literature [88, 90, 37] that use GNNs directly as the backbones for graph AL, our Bayesian probabilistic learning framework naturally supports the dynamic process of AL and allows for seamless updates of the model as new labeled data become available, thus eliminating the need for full model re-training during node selection. This key advantage of our Bayesian probabilistic learning framework paves the way for efficient and interpretable design of our subsequent acquisition function. Following the principle of Expected Model Change Maximization (EMCM) [6] in general AL, we can adjust the likelihood and efficiently update posterior beliefs about the expected change in model parameters, assuming the label of a candidate node from the pool becomes available. More discussions are found in Appendix C.

### 3.1 A Bayesian Probabilistic Learning Framework for GNNs under Semi-supervised Setting

#### 3.1.1 Optimization View of SGC

We choose the SGC model [68] as an example for derivation and present a Bayesian probabilistic learning framework for the semi-supervised node classification. Recall that in Sec. 2.2, SGC aims to solve the following optimization problem Eq. (2) for semi-supervised node classification.

$$\min_{\boldsymbol{\Theta}} \sum_{i \in \mathcal{V}_l} \ell(\mathsf{S}(\mathbf{U}_i(\boldsymbol{\Theta})), \boldsymbol{y}_i), \quad \text{s.t.} \quad \mathbf{U}(\boldsymbol{\Theta}) = \hat{\tilde{\mathbf{A}}}^K \mathbf{X} \boldsymbol{\Theta}. \tag{2}$$

The objective function in Eq. (2) characterizes the backpropagation process, while the constraint in Eq. (2) describes forward propagation. Different GNNs typically exhibit unique forms of node embeddings $\mathbf{U}(\boldsymbol{\Theta})$ in Eq. (2). Notably, several recent studies [42, 102] have focused on unifying the forward pass of GNNs within an optimization framework, but these approaches neglect the backward pass for GNN training. Some research works integrate label information into the unifying framework for GNNs [84, 74], yet none of them provide clear Bayesian interpretations. However, we can still transform the constraint in Eq. (2) into another optimization subproblem [102] based on Theorem 3.1.

**Theorem 3.1.** *The forward pass of a $K$-layer SGC, $\mathbf{U}(\mathbf{\Theta}) = \hat{\tilde{\mathbf{A}}}^K \mathbf{X} \mathbf{\Theta}$, optimizes the following problem by performing $K$ steps of gradient descent:*

$$\hat{\mathbf{\Theta}}^* = \arg\min_{\hat{\mathbf{\Theta}}} \mathrm{Tr}(\mathbf{U}^\intercal \tilde{\mathbf{L}} \mathbf{U}), \quad s.t. \quad \mathbf{U} = \mathbf{X}\hat{\mathbf{\Theta}}.$$

*Here, $\hat{\mathbf{\Theta}}$ is initialized as $\hat{\mathbf{\Theta}}^{(0)} = \mathbf{\Theta}$ and $\tilde{\mathbf{L}} = \mathbf{I} - \hat{\tilde{\mathbf{A}}}$ is the normalized Laplacian matrix. Then,*

$$\mathbf{U}(\mathbf{\Theta}) = \hat{\tilde{\mathbf{A}}}^K \mathbf{X} \mathbf{\Theta} = \mathbf{X} \hat{\mathbf{\Theta}}^*$$

Therefore, Eq. (2) now becomes a bi-level optimization problem as Eq. (3).

$$\min_{\mathbf{\Theta}} \sum_{i \in \mathcal{V}_l} \ell(\mathsf{S}(\mathbf{U}_i(\mathbf{\Theta})), \boldsymbol{y}_i), \quad \text{s.t.} \quad \mathbf{U}(\mathbf{\Theta}) = \mathbf{X}\hat{\mathbf{\Theta}}^*, \hat{\mathbf{\Theta}}^* = \arg\min_{\hat{\mathbf{\Theta}}, \mathbf{U} = \mathbf{X}\hat{\mathbf{\Theta}}, \hat{\mathbf{\Theta}}^{(0)} = \mathbf{\Theta}} \mathrm{Tr}(\mathbf{U}^\intercal \tilde{\mathbf{L}} \mathbf{U}). \quad (3)$$

### 3.1.2 From Optimization View to Bayesian Probabilistic View

Although Eq. (3) is an equivalent optimization view of SGC for semi-supervised node classification, it does not possess a clear Bayesian interpretation. The upper-level objective in Eq. (3) minimizes the supervision loss, and the lower-level constraint $\mathrm{Tr}(\mathbf{U}^\intercal\tilde{\mathbf{L}}\mathbf{U})$ minimizes the graph regularization loss, promoting the homophily assumption that connected nodes share the similar embeddings/labels. Therefore, motivated by [23], we absorb the lower-level constraint into the objective function itself as a relaxed single-level optimization problem in Eq. (4) by minimizing both two terms simultaneously.

$$\min_{\substack{\mathbf{\Theta} \\ \mathbf{U}(\mathbf{\Theta}) = \mathbf{X}\mathbf{\Theta}}} \mathcal{L} = \sum_{i \in \mathcal{V}_l} \ell(\mathsf{S}(\mathbf{U}_i(\mathbf{\Theta})), \boldsymbol{y}_i) + \mathrm{Tr}\left(\mathbf{U}(\mathbf{\Theta})^\intercal \tilde{\mathbf{L}}_\lambda \mathbf{U}(\mathbf{\Theta})\right), \quad (4)$$

where $\tilde{\mathbf{L}}_\lambda = \tilde{\mathbf{L}} + \lambda\mathbf{I}$ with $\lambda$ as the balancing factor between two terms. The proposed optimization problem Eq. (4) immediately leads to a Bayesian probabilistic interpretation [67]. For notational simplicity, we focus on the detailed derivation of the binary case in the main body. In the special case of binary node classification, $\mathbf{U}(\mathbf{\Theta}) \in \mathbb{R}^{n \times c}$ can be reduced to $\mathbf{u}(\mathbf{\Theta}) \in \mathbb{R}^n$, and $\mathsf{S}(\cdot)$ now becomes the Sigmoid function $\sigma(\cdot)$ in Eq. (5).

$$\min_{\substack{\mathbf{\Theta} \\ \mathbf{u}(\mathbf{\Theta}) = \mathbf{X}\mathbf{\Theta}}} \mathcal{L} = \sum_{i \in \mathcal{V}_l} \ell(\sigma(u_i(\mathbf{\Theta})), y_i) + \mathrm{Tr}\left(\mathbf{u}(\mathbf{\Theta})^\intercal \tilde{\mathbf{L}}_\lambda \mathbf{u}(\mathbf{\Theta})\right), \quad (5)$$

Note that now $u_i(\mathbf{\Theta}) \in \mathbb{R}$ and $y_i \in \{0, 1\}$. Then Theorem 3.2 explicitly reveals the Bayesian probabilistic interpretation of the optimization framework in Eq. (5) [67].

**Theorem 3.2.** *Solving Eq. (5) is equivalent to finding the maximum a posteriori (MAP) estimate of the posterior probability distribution with the density $\mathbb{P}(\mathbf{u}(\mathbf{\Theta}) \mid \mathbf{y})$ as*

$$\mathbb{P}(\mathbf{u}(\mathbf{\Theta}) \mid \mathbf{y}) \propto \mu(\mathbf{u}(\mathbf{\Theta})) \exp\left(-\Phi_\ell(\mathbf{u}(\mathbf{\Theta}))\right). \quad (6)$$

*Here, the prior $\mu(\mathbf{u}(\mathbf{\Theta}))$ follows a Gaussian prior $\mathcal{N}(\mathbf{0}, \tilde{\mathbf{L}}_\lambda^{-1})$ and the likelihood $\exp\left(-\Phi_\ell(\mathbf{u}(\mathbf{\Theta}))\right)$ is defined by the likelihood potential $\Phi_\ell(\mathbf{u}(\mathbf{\Theta})) := \sum_{i \in \mathcal{V}_l} \ell(\sigma(u_i(\mathbf{\Theta})), y_i)$. $\mathbf{y} = [y_i]_{i \in \mathcal{V}_l} \in \{0, 1\}^{n_l}$.*

The Gaussian prior denotes a prior belief over the distribution of the node embedding $\mathbf{u}(\mathbf{\Theta})$ governed by the graph structure or captured by the Laplacian matrix in the covariance term. It relates to the second trace term in Eq. (5) or the forward pass of SGC. The likelihood ($\mathbb{P}(\mathbf{y} \mid \mathbf{u}(\mathbf{\Theta}))$) represents the underlying assumptions about how the observed labels $y_i$ are generated, determined by $\ell(\cdot, \cdot)$. It relates to the first supervision loss term in Eq. (5) or the backward pass for training SGC.

Theorem 3.2 provides a clear Bayesian interpretation of SGC for semi-supervised node classification. The forward pass defines a prior over node embeddings parametrized by $\mathbf{\Theta}$, while the backward pass specifies the likelihood based on the supervision loss function. Together, they yield the posterior, guiding the update of node embeddings after observing limited labels. This insight enables the utilization of the EMCM method [6], which selects nodes leading maximum model change. For efficiency in query node selection, we focus on SGC in this work, despite more complex GNNs like GCN yielding a non-Gaussian prior in Eq. (6). The analysis of more complex GNNs, such as GCN, yields a non-Gaussian prior while keeping all other aspects unchanged in Theorem 3.2. However, for computational efficiency in subsequent query node selection, we adhere to the SGC model for now.

### 3.2 Expected Model Change Maximization for Graph Active Learning

#### 3.2.1 Challenges for Applying EMCM on Graphs

We choose the EMCM method [6] for three main reasons. First, it is a performance-based approach that directly assesses the expected impact on the model if a sample is selected for a query, aligning perfectly with the goal of AL. Second, it offers lower computational costs compared to other performance-based methods like EER. Third, it is ideally supported by our proposed Bayesian probabilistic learning framework which provides an efficient way to estimate the expected model change without re-training. In the sequential active learning setting, our objective is to select the most informative node $k^* = \arg\max_{k \in \mathcal{V}_u} \mathcal{A}(k)$ with a designed acquisition function $\mathcal{A}(\cdot)$. Similar to other performance-based methods, EMCM utilizes a *look-ahead* model [31] with the modified objective as $\mathcal{L}^{k,y_k^+} = \mathcal{L} + \ell(\sigma(u_k(\boldsymbol{\Theta})), y_k^+)$ with $\mathcal{L}$ from Eq. (5), where we add the unlabeled node $k \in \mathcal{V}_u$ with the *hypothetical label* $y_k^+ \in \{0, 1\}$ due to the unavailability of the true label $y_k$. We select the node that could maximally change the node embeddings if it is added to the query set, via

$$k^* = \arg\max_{k \in \mathcal{V}_u} \mathcal{A}(k) = \arg\max_{k \in \mathcal{V}_u} \sum_{y_k^+ \in \{0,1\}} \mathbb{P}(y_k^+ \mid k) \|\mathbf{u}(\tilde{\boldsymbol{\Theta}}^*) - \mathbf{u}(\boldsymbol{\Theta}^*)\|_2. \tag{7}$$

Here, $\tilde{\boldsymbol{\Theta}}^* = \arg\min \mathcal{L}^{k,y_k^+}$, $\boldsymbol{\Theta}^* = \arg\min \mathcal{L}$ and $\mathbb{P}(y_k^+ \mid k)$ is the predicted probability of label $y_k^+$ for node $k$ estimated by the current model. The high-level motivation of Eq. (7) is that the generalization error changes only when the current model is updated. Nodes that do not update the node embeddings are useless for AL. Nodes causing significant model changes in terms of node embeddings are expected to lead to faster convergence to the optimal model. More detailed illustrations of EMCM can be found in Appendices B.3 and C.2.1.

Re-training the look-ahead model for the fixed node $k$ in Eq. (7) to obtain the exact solution is computationally infeasible due to the large number of unlabeled nodes. However, the proposed equivalent Bayesian learning framework Eq. (6) offers an efficient alternative. By utilizing rank-one updates of the current model's posterior mean and covariance (Eq. (5)), the posterior mean and covariance of the look-ahead model can be computed efficiently without the need for re-training. It significantly reduces the computational cost involved in applying the EMCM method on graphs.

#### 3.2.2 Truncated Spectral Projection and Laplace Approximation for Efficiency

The computation of $\mathbf{u}(\boldsymbol{\Theta}^*)$ or $\mathbf{u}(\tilde{\boldsymbol{\Theta}}^*)$ through the Bayesian learning method in Eq. (6) still requires storing a large covariance matrix $\tilde{\mathbf{L}}_\lambda^{-1} \in \mathbb{R}^{n \times n}$. Motivated by spectral clustering [48], we utilize the first $m$ ($m \ll n$) smallest eigenvalues of $\tilde{\mathbf{L}}$ and their corresponding eigenvectors that contain important geometric information of the graph. Namely, we introduce $\boldsymbol{\Lambda}_\lambda = \text{diag}(\lambda_1 + \lambda, \cdots, \lambda_m + \lambda)$ and $\mathbf{V} = [\mathbf{v}^1, \cdots, \mathbf{v}^m] \in \mathbb{R}^{n \times m}$, where $\mathbf{v}^i$ is the eigenvector corresponding to the $i$-th eigenvalue $\lambda_i$. We can now project the node embedding $\mathbf{u}(\boldsymbol{\Theta})$ onto the space spanned by these $m$ eigenvectors by $\boldsymbol{\alpha} = \mathbf{V}^\intercal \mathbf{u}(\boldsymbol{\Theta})$. Based on the orthogonality of $\mathbf{V}$ ($\mathbf{V}^\intercal \mathbf{V} = \mathbf{I}$), we can easily convert Eq. (5) as,

$$\min_{\boldsymbol{\alpha}} \tilde{\mathcal{L}} = \sum_{i \in \mathcal{V}_l} \ell(\sigma(\mathbf{e}_i^\intercal \mathbf{V} \boldsymbol{\alpha}), y_i) + \text{Tr}\left(\boldsymbol{\alpha}^\intercal \boldsymbol{\Lambda}_\lambda \boldsymbol{\alpha}\right). \tag{8}$$

Eq. (8) now restricts the model latent space from $\mathbb{R}^n$ to $\mathbb{R}^m$, speeding up the model training and reducing the spatial complexity. A similar Bayesian interpretation of Eq. (8) regarding $\boldsymbol{\alpha} \in \mathbb{R}^m$ can be trivially extended from Theorem 3.2. We term this technique as truncated spectral projection.

To further reduce the computational overhead, we apply Laplace approximation on the posterior distribution of $\mathbb{P}(\boldsymbol{\alpha} \mid \mathbf{y})$ (instead of $\mathbb{P}(\mathbf{u}(\boldsymbol{\Theta}) \mid \mathbf{y})$) corresponding to Eq. (6). The key idea is to approximate the non-Gaussian posterior via suitable Gaussian distributions here so as to obtain the analytical closed-form of the look-ahead model's posterior mean and covariance later [47].

**Theorem 3.3.** *The Laplace approximation of the posterior distribution regarding* $\boldsymbol{\alpha} = \mathbf{V}^\intercal \mathbf{u}(\boldsymbol{\Theta}) \in \mathbb{R}^m$ *according to Eq. (8) and Eq. (6) is given as follows.*

$$\boldsymbol{\alpha} \mid \mathbf{y} \sim \mathcal{N}(\hat{\boldsymbol{\alpha}}, \hat{\mathbf{C}}_{\hat{\boldsymbol{\alpha}}}), \hat{\boldsymbol{\alpha}} = \arg\min_{\boldsymbol{\alpha}} \tilde{\mathcal{L}}, \hat{\mathbf{C}}_{\hat{\boldsymbol{\alpha}}} = (\boldsymbol{\Lambda}_\lambda + \mathbf{V}^\intercal (\sum_{i \in \mathcal{V}_l} F'(\sigma(\mathbf{e}_i^\intercal \mathbf{V} \hat{\boldsymbol{\alpha}}), y_i) \mathbf{e}_i \mathbf{e}_i^\intercal) \mathbf{V})^{-1}. \tag{9}$$

*Here, we define* $F'(x, y) := \frac{\partial^2 \ell}{\partial x^2}(x, y)$. $\mathbf{e}_i$ *represents the $i$-th standard basis vector.*

Theorem 3.3 gives a closed-form approximated Gaussian posterior distribution of $\boldsymbol{\alpha}$. Note that $\hat{\boldsymbol{\alpha}} = \mathbf{V}^\mathsf{T}\mathbf{u}(\boldsymbol{\Theta}^*)$ denotes the *current* model's $\tilde{\mathcal{L}}$ MAP estimator in Eq. (8) based on Theorem 3.2.

### 3.2.3 Expected Model Change Acquisition Function

Following the definition of $\mathcal{L}^{k,y_k^+}$ over $\boldsymbol{\Theta}$, the look-ahead model now optimizes $\tilde{\mathcal{L}}^{k,y_k^+} = \tilde{\mathcal{L}} + \ell(\sigma(\mathbf{e}_i^\mathsf{T}\mathbf{V}\boldsymbol{\alpha}), y_k^+)$ with $\tilde{\mathcal{L}}$ in Eq. (8). We use $\hat{\boldsymbol{\alpha}}^{k,y_k^+}$ to denote the look-ahead model's $\tilde{\mathcal{L}}^{k,y_k^+}$ MAP estimator. The exact value of $\hat{\boldsymbol{\alpha}}^{k,y_k^+}$ can be obtained by re-training the look-ahead model. We instead compute an approximation of $\hat{\boldsymbol{\alpha}}^{k,y_k^+}$ by one single step of Newton's method on the objective $\tilde{\mathcal{L}}^{k,y_k^+}$ without re-training, given by Theorem 3.4. Since the look-ahead model is built upon the current model, using a single step of Newton's Method can quickly approximate the solution for $\tilde{\mathcal{L}}^{k,y_k^+}$ based on the local information (gradient and curvature) around the current model's MAP estimator [47].

**Theorem 3.4.** *The look-ahead MAP estimator $\hat{\boldsymbol{\alpha}}^{k,y_k^+}$ can be obtained by performing one step of Newton's method on $\tilde{\mathcal{L}}^{k,y_k^+}$ from the current MAP estimator $\hat{\boldsymbol{\alpha}}$ as*

$$\hat{\boldsymbol{\alpha}}^{k,y_k^+} = \hat{\boldsymbol{\alpha}} - \frac{F(\sigma(\mathbf{e}_k^\mathsf{T}\mathbf{V}\hat{\boldsymbol{\alpha}}), y_k^+)}{1 + F'(\sigma(\mathbf{e}_k^\mathsf{T}\mathbf{V}\hat{\boldsymbol{\alpha}}), y_k^+)(\mathbf{e}_k^\mathsf{T}\mathbf{V})\hat{\mathbf{C}}_{\hat{\boldsymbol{\alpha}}}(\mathbf{e}_k^\mathsf{T}\mathbf{V})^\mathsf{T}} \hat{\mathbf{C}}_{\hat{\boldsymbol{\alpha}}}(\mathbf{e}_k^\mathsf{T}\mathbf{V})^\mathsf{T} \tag{10}$$

*We define $F(x,y) \coloneqq \frac{\partial \ell}{\partial x}(x,y)$, $F'(x,y) \coloneqq \frac{\partial^2 \ell}{\partial x^2}(x,y)$. $\mathbf{e}_i$ represents the $i$-th standard basis vector.*

Now, we finally come back to the original acquisition function for applying EMCM on graphs in Eq. (7). The predicted probability of label $y_k^+$ can be set as the soft label prediction for node $k$ by the current model $\mathbb{P}(y_k^+ = 1 \mid k) = \sigma(\mathbf{e}_k^\mathsf{T}\mathbf{u}(\boldsymbol{\Theta}))$ $(\mathbb{P}(y_k^+ = 0 \mid k) = 1 - \sigma(\mathbf{e}_k^\mathsf{T}\mathbf{u}(\boldsymbol{\Theta})))$. Thanks to the truncated spectral projection ($\hat{\boldsymbol{\alpha}} = \mathbf{V}^\mathsf{T}\mathbf{u}(\boldsymbol{\Theta}^*)$, $\hat{\boldsymbol{\alpha}}^{k,y_k^+} = \mathbf{V}^\mathsf{T}\mathbf{u}(\tilde{\boldsymbol{\Theta}}^*)$) and Laplace approximation, we can now measure the change of $\|\mathbf{u}(\tilde{\boldsymbol{\Theta}}^*) - \mathbf{u}(\boldsymbol{\Theta}^*)\|_2$ via $\|\hat{\boldsymbol{\alpha}}^{k,y_k^+} - \hat{\boldsymbol{\alpha}}\|_2$ due to the orthogonal invariance of $l_2$ norm. Based on Theorem 3.4 and Eq. (7), the final designed acquisition function is

$$\mathcal{A}(k) = \sum_{y_k^+ \in \{0,1\}} \mathbb{P}(y_k^+ = 1|k)^{y_k^+} \mathbb{P}(y_k^+ = 0|k)^{1-y_k^+} \left| \frac{F(\sigma(\mathbf{v}_k^\mathsf{T}\hat{\boldsymbol{\alpha}}), y_k^+)}{1 + F'(\sigma(\mathbf{v}_k^\mathsf{T}\hat{\boldsymbol{\alpha}}), y_k^+)\mathbf{v}_k^\mathsf{T}\hat{\mathbf{C}}_{\hat{\boldsymbol{\alpha}}}\mathbf{v}_k} \right| \|\hat{\mathbf{C}}_{\hat{\boldsymbol{\alpha}}}\mathbf{v}_k\|_2. \tag{11}$$

Here, $\mathbb{P}(y_k^+ = 1|k) = \sigma(\mathbf{e}_k^\mathsf{T}\mathbf{u}(\boldsymbol{\Theta}))$ and define $\mathbf{v}_k \coloneqq (\mathbf{e}_k^\mathsf{T}\mathbf{V})^\mathsf{T}$ as the $k$-th row vector of $\mathbf{V}$. The proposed active learning algorithm expecte**D** m**O**del **C**hange maximiza**T**ion **O**n g**R**aphs (DOCTOR) in the sequential setting is presented in Algorithm 2 in Appendix D.2. For the batch active learning setting and the multi-class classification setting, we refer to Appendices D.3 and D.4.

## 3.3 Theoretical Insights of the Proposed Method DOCTOR

We analyze the profound connection between the proposed EMCM method and the Expected Error Minimization (EEM) or EER method for graph active learning. We will show that the designed acquisition function Eq. (11) will be reduced to the one for EEM under some assumptions and thus it has a direct connection to the ultimate goal for graph AL in Eq. (1). For clarity, we focus on the fundamental regression task within the sequential AL setting. We choose the regression for theoretical analysis to get rid of the non-linear activation functions in the Bayesian learning framework ($\sigma(\cdot)$ in Eq. (5)) from the beginning. We also fix the loss function $\ell(\cdot, \cdot)$ as the common squared error loss.

**Theorem 3.5.** *We assume $\ell(x,y) = \frac{1}{2}(x-y)^2$ is used for the regression task in the sequential active learning setting (b=1). In the $t$-th iteration step, the proposed DOCTOR algorithm aims to solve*

$$\max_{k \in \mathcal{V}_u^{(t)}} \mathcal{A}(k) = \max_{k \in \mathcal{V}_u^{(t)}} \frac{1}{1 + \mathbf{v}_k^\mathsf{T}\hat{\mathbf{C}}_{\hat{\boldsymbol{\alpha}}}} \|\hat{\mathbf{C}}_{\hat{\boldsymbol{\alpha}}}\mathbf{v}_k\|_2^2. \tag{12}$$

*And the Expected Error Minimization on graphs aims to solve $\min_{k \in \mathcal{V}_u^{(t)}} \mathbb{E}_{i \in \mathcal{V}_u^{(t)} \setminus \{k\}}[\ell(\hat{y}_i, y_i)]$. Then,*

$$\underset{k \in \mathcal{V}_u^{(t)}}{\arg\min} \, \mathbb{E}_{i \in \mathcal{V}_u^{(t)} \setminus \{k\}}[\ell(\hat{y}_i, y_i)] = \underset{k \in \mathcal{V}_u^{(t)}}{\arg\max} \, \frac{1}{1 + \mathbf{v}_k^\mathsf{T}\hat{\mathbf{C}}_{\hat{\boldsymbol{\alpha}}}} \|\hat{\mathbf{C}}_{\hat{\boldsymbol{\alpha}}}\mathbf{v}_k\|_2^2. \tag{13}$$

Table 1: Test accuracy (%) on five datasets with the same labeling budget (20 labels per class).

| Method | Cora | Citeseer | PubMed | Reddit | ogbn-arxiv |
|---|---|---|---|---|---|
| Random | 78.9±0.9 | 70.7±0.7 | 78.4±0.5 | 91.3±0.5 | 68.3±0.4 |
| AGE | 82.5±0.5 | 71.4±0.5 | 79.4±0.7 | 91.5±0.4 | 68.7±0.3 |
| ALG | 82.6±0.6 | 73.6±0.6 | 80.8±0.3 | 92.4±0.6 | 70.0±0.2 |
| RIM | 84.1±0.8 | 73.2±0.7 | 80.2±0.4 | 92.0±0.7 | 70.5±0.8 |
| IGP | 86.3±0.7 | 75.8±0.4 | 83.5±0.5 | 93.3±0.4 | 70.9±0.5 |
| GraphPart | 86.5±1.2 | 74.0±2.0 | 81.5±1.6 | 92.7±1.0 | 72.3±2.1 |
| DOCTOR | **86.9±0.7** | **76.5±0.8** | **84.3±0.9** | **93.5±0.5** | **73.0±1.2** |

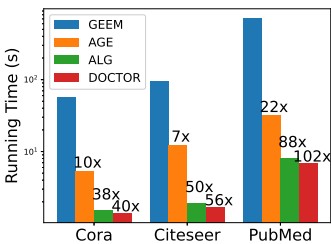

Figure 1: Running Time Comparison (at log scale).

Theorem 3.5 demonstrates the equivalence between the proposed DOCTOR algorithm and the EEM method on graphs, given certain basic assumptions. Consequently, from a theoretical standpoint, DOCTOR can implicitly select the node that minimizes the expected prediction error if added into the training set during the next iteration, as directly correlated with the initial objective in Eq. (1). Moreover, DOCTOR offers significantly reduced computational overhead through truncated spectral projection and Laplace approximation, as compared to EEM-based approaches like GEEM [51].

## 4 Experiments

We now verify the effectiveness of DOCTOR on five real-world graphs. We aim to evaluate DOCTOR in terms of prediction accuracy, efficiency, generalization ability, and interpretability (Appendix G).

### 4.1 Experimental Setup

**Datasets** We evaluate DOCTOR for the pool-based active learning setting on five datasets, including three citation networks (i.e., **Citeseer**, **Cora** and **PubMed**) [32], one social network (**Reddit**) [22], and one large-scale OGB dataset (**ogbn-arxiv**) [26]. Details can be found in Appendix F.1.

**Baselines** We compare DOCTOR with six state-of-the-art baseline methods for active learning on graphs. **Random** selects the nodes to query randomly. **AGE** [4] Combines different query strategies linearly with time-sensitive parameters. **ALG** [88] select nodes that maximize the effective reception field in GCN. **RIM** [89] further considers the influence quality during node selection. **IGP** [90] choose nodes that maximize information gain propagation. **GraphPart** [41] selects representative nodes within each graph partition to query. Implementation details are referred to in Appendix F.2.

### 4.2 Experimental Results

**Accuracy Comparison** We compare DOCTOR with baselines for multi-class node classification in the batch AL setting (Appendix D.4). We choose a small set of two randomly sampled labeled nodes for each method as the initial pool. The query node size is fixed as $b = 5$ in each iteration and the labeling budget is fixed as 20 labels per class ($B = 20c$). The backbone is set as the SGC model. Table 1 shows the classification accuracy on five datasets with the same labeling budget. We repeat each method 10 times and report the mean and variance regarding the accuracy. Remarkably, the proposed DOCTOR achieves the best in all the cases and improves the second-best baseline by nearly 1% on some datasets. The improvement on the Reddit dataset is rather marginal since it is designed for the inductive setting and DOCTOR is originally proposed for the transductive setting.

To show the influence of the labeling budget, we display the accuracy of different AL methods under different labeling budgets on three citation datasets in Figure 2. We range the labeling budgets $B$ from $2c$ to $20c$ with $c$ as the number of classes. The results in Figure 2 demonstrate that with the increase in labeling budget, the accuracy of DOCTOR grows as well, outperforming many of the baselines with a greater margin.

**Efficiency Comparison** To evaluate the efficiency of DOCTOR, we conducted an analysis of its running time per AL iteration alongside other baselines using the Cora dataset. We introduce another performance-based graph AL method called GEEM [51], which extends the EEM principle

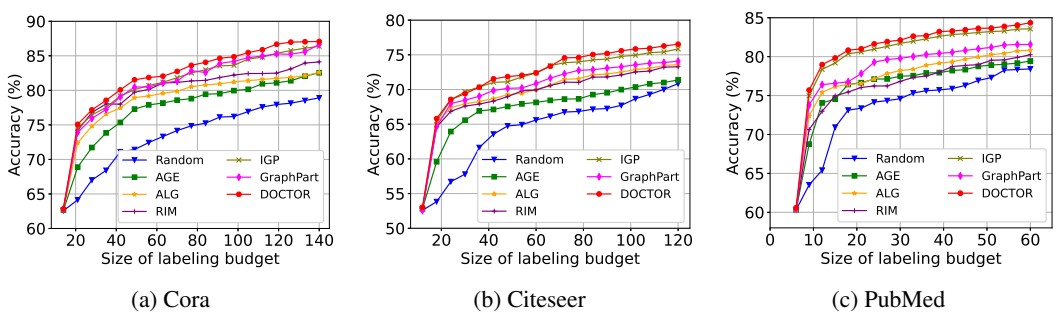

| (a) Cora | (b) Citeseer | (c) PubMed |

Figure 2: Test accuracy with different labeling budgets on three datasets.

Table 2: Test accuracy (%) of different methods with different backbone GNNs on the Citeseer dataset.

| Method | SGC | GCN | APPNP | GraphSAGE |
|---|---|---|---|---|
| Random | 70.7±0.7 | 70.8±0.8 | 71.0±0.6 | 70.6±0.8 |
| AGE | 71.4±0.5 | 71.6±0.6 | 71.3±0.5 | 71.4±0.7 |
| ALG | 73.6±0.6 | 73.8±0.5 | 74.1±0.7 | 74.0±0.9 |
| RIM | 73.2±0.7 | 73.0±0.8 | 73.3±0.8 | 73.2±0.9 |
| IGP | 75.8±0.4 | 75.4±0.9 | 75.5±0.6 | 75.6±1.0 |
| GraphPart | 74.0±2.0 | 74.3±1.6 | 74.5±1.3 | 74.9±1.5 |
| DOCTOR | **76.5±0.8** | **76.2±0.7** | **76.6±0.9** | **76.4±0.9** |

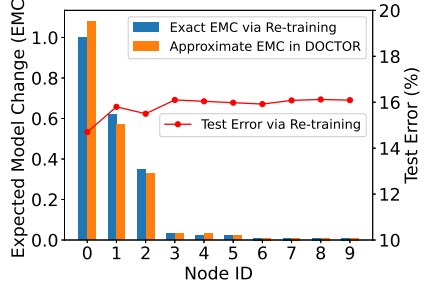

Figure 3: Interpretation Analysis.

for graphs. However, GEEM's scalability is extremely limited on large-scale datasets. We consider the training time of GEEM as the baseline and measure the relative running time of each method. Figure 1 illustrates these results, where the numbers atop each bar represent the multiplier improvement in running time compared to GEEM. Our method, DOCTOR, exhibits significant efficiency improvements, surpassing GEEM dozens of times while achieving comparable results to ALG. The more detailed time complexity of the proposed DOCTOR method can be found in Appendix G.1.

**Generalization to Other GNNs** Note that the acquisition function in the proposed DOCTOR algorithm is specifically designed for the SGC method [68]. Nonetheless, we can still substitute the backbone training model, SGC, with other GNNs in each iteration of the AL loop (line 5 in Algorithm 2) without any difficulties. The resulting acquisition function will be computed based on the new node embeddings generated by other GNNs without any other changes. We select some common GNNs, including GCN [32], APPNP [33] and GraphSAGE [22], to verify the generalization ability of the proposed acquisition function. We change the backbone model of other baselines accordingly as well. We fix the labeling budget as 20 labels per class. All experiments are conducted in the transductive AL setting on the Citeseer dataset and the results are summarized in Table 2. Like other baselines, the proposed acquisition function in the proposed DOCTOR method generalizes well to other GNN backbone models without any significant changes regarding prediction accuracy, even though it is originally designed for SGC. The underlying reason is that apart from SGC, most other GNNs can also be incorporated into the proposed Bayesian learning framework with more complex forms of prior and the currently derived form of the acquisition function can approximate the exact solution when other GNNs are used as backbones.

**Interpretability** We perform a case study on a single iteration step of DOCTOR in the sequential AL setting using the Cora dataset. We choose one query node from a pool of 10 randomly sampled unlabeled nodes for visualization purposes. We focus on one iteration step in the exact middle of the entire AL process. We compare the approximate EMC, computed based on Eq. (11) in the DOCTOR method, with the exact EMC, computed based on Eq. (7) by adding the node's label for re-training. We use the maximum value of the exact EMC among these 10 nodes as the baseline, and we report all other EMC values in a relative manner in Figure 3. The results show that our efficient computation of the designed acquisition function via truncated spectral projection and Laplace approximation approximates the exact solution quite well. Additionally, we sort the candidate nodes in descending

order based on the exact EMC value and obtain the respective test error by re-training the model with the node's label. Figure 3 illustrates that by choosing the node that has the maximum approximate EMC value without re-training via the proposed acquisition function Eq. (11), we can minimize the test error on the remaining unlabeled nodes by adding this node to the labeled node set for the next round of training. This key observation empirically validates the insights of Theorem 3.5.

## 5 Conclusion

In this work, we extend the EMCM principle for GNNs based on a provided Bayesian learning framework. This allows us to efficiently compute a closed-form acquisition function that can be used to select the most informative nodes to label. The proposed DOCTOR method establishes theoretical connections with the ultimate goal of minimizing expected prediction error in AL. This makes it a promising tool for training GNNs in real-world applications with limited labeled data.

## 6 Limitations

The acquisition function in DOCTOR is derived from the SGC model but has good generalizability to other GNNs empirically. We leave the investigation of the closed-form MAP estimator of other GNNs as future work. The acquisition function in DOCTOR is originally designed for the sequential active learning setting, and we only use a simple sampling method to extend it for the batch active learning setting. We also leave the investigation of a more advanced batch AL extension of DOCTOR as future work. The theoretical connection between our method with entropy minimization in terms of strictly proper scoring rules can be further investigated in detail [64]. Other potential applications of our method can be explored in the hyperbolic space [76, 79, 77, 80, 78, 75, 81] or in the natural language processing domain [34, 36, 35, 20, 19, 63, 44, 99, 100, 98].

## Acknowledgements

The work described in this paper was partially supported by the National Key Research and Development Program of China (No. 2018AAA0100204), RGC Research Impact Fund (RIF), R5034-18 (CUHK 2410021), and RGC General Research Funding Scheme (GRF) 14222922 (CUHK 2151185).

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
