# A Notation Table

We summarize the frequently-used notations in this paper in Table 3.

Table 3: Notations used in this paper.

| Notation | Definition |
|---|---|
| $n$ | Number of nodes |
| $n_l$ | Number of labeled nodes |
| $n_u$ | Number of unlabeled nodes |
| $k$ | Node index |
| $c$ | Number of classes |
| $d$ | Dimension of node features |
| $b$ | Size of query node set |
| $B$ | Total labeling budget |
| $m$ | Number of smallest eigenvalues of $\tilde{\mathbf{L}}$ |
| $\mathcal{G}$ | Graph |
| $\mathcal{V}$ | Node (index) set |
| $\mathcal{E}$ | Edge Set |
| $\mathcal{V}_l$ | Labeled node (index) set |
| $\mathcal{V}_u$ | Unlabeled node (index) set |
| $\mathcal{V}_q$ | Query node (index) set |
| $\mathbf{X} \in \mathbb{R}^{n \times d}$ | Node feature matrix |
| $\mathbf{A} \in \mathbb{R}^{n \times n}$ | Unormalized adjacency matrix |
| $\tilde{\mathbf{A}} \in \mathbb{R}^{n \times n}$ | Unormalized adjacency matrix with added self-loop |
| $\hat{\tilde{\mathbf{A}}} \in \mathbb{R}^{n \times n}$ | Normalized adjacency matrix with added self-loop |
| $\mathbf{D} \in \mathbb{R}^{n \times n}$ | Degree matrix |
| $\tilde{\mathbf{D}} \in \mathbb{R}^{n \times n}$ | Degree matrix with added self-loop |
| $\tilde{\mathbf{L}} \in \mathbb{R}^{n \times n}$ | Normalized Laplacian matrix with added self-loop |
| $\tilde{\mathbf{L}}_\lambda \in \mathbb{R}^{n \times n}$ | $\tilde{\mathbf{L}} + \lambda \mathbf{I}$ |
| $\boldsymbol{\Theta}$ | Trainable parameters |
| $\mathbf{U}(\boldsymbol{\Theta}) \in \mathbb{R}^{n \times c}$ | Node embedding matrix (before softmax function) |
| $\mathbf{U}_i(\boldsymbol{\Theta}) \in \mathbb{R}^{c}$ | Node embedding vector for node $i$ |
| $\mathbf{u}(\boldsymbol{\Theta}) \in \mathbb{R}^{n}$ | Node embedding matrix for binary classification |
| $u_i(\boldsymbol{\Theta}) \in \mathbb{R}$ | Node embedding vector for node $i$ for binary classification |
| $\boldsymbol{\Lambda}_\lambda \in \mathbb{R}^{m \times m}$ | $m$ smallest eigenvalues of $\tilde{\mathbf{L}}_\lambda$ |
| $\mathbf{V} = [\mathbf{v}^1, \cdots, \mathbf{v}^m] \in \mathbb{R}^{n \times m}$ | $\mathbf{v}^i$ is the eigenvector corresponding to $\lambda_i$ |
| $\boldsymbol{\alpha} = \mathbf{V}^\intercal \mathbf{u}(\boldsymbol{\Theta}) \mathbb{R}^m$ | Node embeddings projected onto the truncated spectral subspace |
| $\boldsymbol{x}_i \in \mathbb{R}^d$ | Node feature for node $i$ |
| $\boldsymbol{y}_i \in \{0,1\}^c$ | Ground-truth hard label for node $i$ |
| $\hat{\boldsymbol{y}}_i \in [0,1]^c$ | Predicted soft label for node $i$ |
| $y_i \in \{0,1\}$ | Groud-truth hard label for node $i$ for binary classification |
| $\hat{y}_i \in [0,1]$ | Predicted soft label for node $i$ for binary classification |
| $\mathsf{S}(\cdot)$ | Softmax function |
| $\sigma(\cdot)$ | Sigmoid function |
| $\phi$ | Activation function |
| $\ell(\cdot, \cdot)\colon \mathbb{R}^c \times \mathbb{R}^c \to \mathbb{R}^+$ | Loss function |

# B Related Work

## B.1 General Active Learning

In numerous fields, acquiring labeled data can be costly. As a result, active learning (AL) [85] has been introduced to develop a classifier that accurately predicts labels for new instances while minimizing the number of training labels required. An AL framework typically comprises two main elements: a query system that selects an instance from the training data to request its label and an oracle that provides the label for the chosen instance. Various algorithms have been suggested by researchers to enhance training performance within a set labeling budget. The majority of the work can be grouped into three categories based on the query strategy [1]: uncertainty-based, representativeness-based, and performance-based. Table 4 and Table 5 contain a detailed comparison of these categories with representative methods. Generally, distinct implementations of the three primary AL categories can be proposed for different classification algorithms. No "optimal" AL solution exists for all classification tasks.

Table 4: Comparison between different general active learning query strategies.

| Categories | Representative Methods | Query Set Selection Strategies |
|---|---|---|
| Uncertainty-based | Uncertainty Sampling [83, 50] | Samples that are most uncertain |
| | Query by Committee (QBC) [56, 52] | Samples that multiple classifiers disagree most |
| Representativeness-based | Density-weighted methods [14, 46, 15] | Samples with the most information density |
| | Cluster-based methods [49, 13, 72] | Samples that are the most representative of clusters |
| Performance-based | Expected Error Reduction (EER) [53] | Samples that lead to the minimum expected prediction error |
| | Expected Variance Reduction [54] | Samples that lead to the minimum expected output variance |
| | Expected Model Change Maximization (EMCM) [6] | Samples that lead to the maximum expected model change |

Table 5: Comparison between different general active learning query strategies (Cont.).

| Categories | Strengths | Weaknesses |
|---|---|---|
| Uncertainty-based | Simple and efficient | Sensitive to outliers |
| Representativeness-based | Robust to outliers | Sensitive to density estimation or clustering algorithms |
| Performance-based | Directly or closely related to the model performance | Computationally expensive |

## B.2 Active Learning on Graphs

We give a brief review of recent works that adapt active learning strategies to graph-structured data. Early works in active learning on graphs [103, 45, 3, 21, 29, 12] are designed specifically for non-deep-learning models and/or fail to take the node features into consideration. As deep geometric learning and GNNs become popular, iterative node selection criteria were designed upon the expressiveness of GNNs. AGE [4] proposes several query node set selection criterion like the information entropy as the uncertainty measure and the density score as information density. ANRMAB [18] improves AGE by introducing a multi-armed bandit mechanism for adaptive decision-making. SEAL [38] devises a novel AL query strategy in an adversarial way. ALG [88] decouples the GNN model and proposes a new node selection metric that maximizes the effective reception field. RIM [89] and IGP [90] both consider selecting the nodes that lead to maximum number of influenced nodes or maximum information gain of all influenced nodes. GRAIN [91] further improves the data efficiency of GNNs via a diversified Influence maximization principle. GraphPart [41] first splits the graph into disjoint partitions and then selects representative nodes within each partition to query. ScatterSample [41] collects nodes with large uncertainty from different regions of the sample space

Table 6: Summarization of different active learning methods for graph-structured data.

| Categories | General Active Learning Strategies | Active Learning on Graphs | |
| --- | --- | --- | --- |
| | | Non-DL-based | DL-based |
| Uncertainty-based | Uncertainty Sampling | [103] | [4, 37, 41] |
| | Query by Committee (QBC) | [3] | [86] |
| Representativeness-based | Density-weighted methods | [69] | [89, 91, 90] |
| | Cluster-based methods | [3] | [41] |
| Performance-based | Expected Error Reduction (EER) | [45, 21] | [51] |
| | Expected Variance Reduction | [29] | - |
| | Expected Model Change Maximization (EMCM) | [47] | Ours |

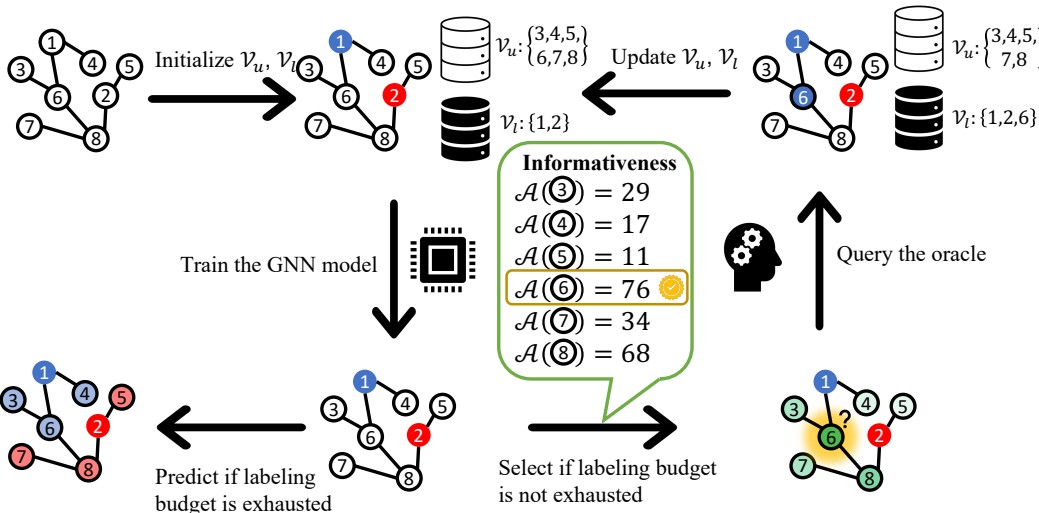

Figure 4: The process of pool-based active learning on graphs (Algorithm 1).

for labeling. SmartQuery [37] uses a hybrid uncertainty reduction function. GEEM [50] extends the Expected Error Reduction method for graphs with a preemptive querying system. Other works also utilize reinforcement learning for AL on graphs [25, 97] or analyze the theoretical guarantees of AL algorithms on graphs [70, 66, 65]. Most of these works extend the general active learning strategies for graph-structured data and we compare some representative methods for active learning on graphs in Table 6.

There are also some existing works that are closely related to our work but are out of the scope of this paper. Instead of the common node classification task, other works also explore active learning strategies for link prediction [30] and graph classification [73]. Some recent works also construct a graph from the training data like images to improve the general active learning algorithms [8, 87]. These works are not comparable with our work since we only focus on active learning for the node-level classification task with the given graph data in hand.

We illustrate the general process of pool-based active learning on graphs in Figure 4 and present the corresponding pseudocode in Algorithm 1. We first initialize a few nodes with the labels and then we enter the active learning loop with the given labeling budget. In each AL interaction, we first train the GNN model in a semi-supervised setting with the existing labeled nodes. This training step is inevitable to obtain meaningful predictions. Next, the most important step is to select the most informative node(s) from the pool of the remaining unlabeled nodes. To measure informativeness, the acquisition function $\mathcal{A}(\cdot)$ of each candidate unlabeled node is usually designed. The node with the largest acquisition function value will be used to query the oracle for its label. Different graph active learning methods reviewed above have different designs of the acquisition function under different principles like EMCM. Finally, we update the labeled node set and the unlabeled node set for the next AL iteration. This process will be repeated until the labeling budget is exhausted and the predictions made by the last trained GNN model are used as the final predictions.

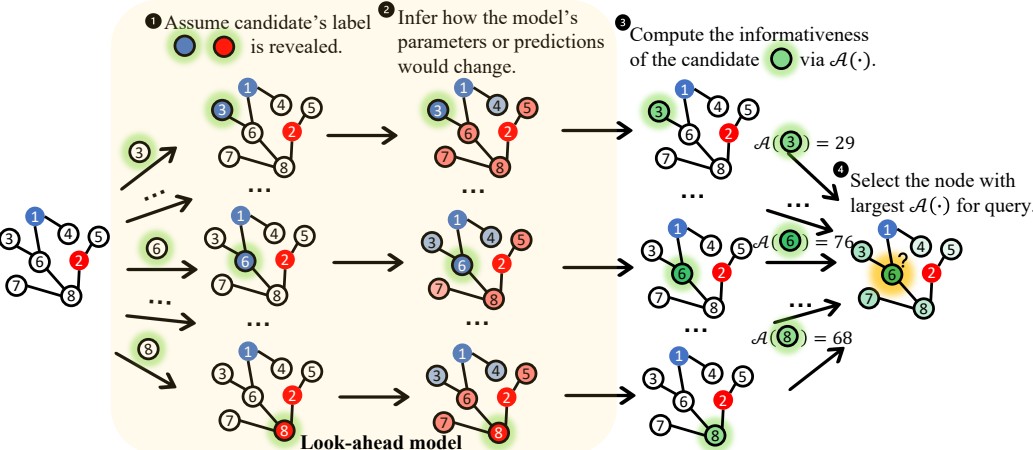

Figure 5: The high-level design principle of performance-based active learning methods on graphs.

## B.3 Expected Model Change Maximization (EMCM)

### B.3.1 Look-ahead Model in Performance-based Methods for Active Learning

Before we introduce the idea of Expected Model Change Maximization (EMCM) in AL, we briefly revisit the design principle of the acquisition function in performance-based methods for active learning, as shown in Figure 5. The performance-based AL methods choose the most informative node for query by analyzing the potential influence of the selected node on the model performance if it is added into the labeled node set for training. More specifically, the performance-based methods usually consider the look-ahead model. We first hypothetically assume one candidate node $\boldsymbol{x}_k^+$ from the pool of the unlabeled nodes is chosen for query and its label is revealed as $\boldsymbol{y}_k^+$ by the oracle. Then we analyze the potential influence of this candidate node $(\boldsymbol{x}_k^+, \boldsymbol{y}_k^+)$ on the model's parameters or predictions if it is added to the labeled node set for the next round of training. The informativeness measurement of this candidate node can then be computed via the designed acquisition function $\mathcal{A}(\cdot)$, recording the change of the model parameters (EMCM) or the prediction error on the remaining unlabeled nodes (EEM/EER). We finally iterate all the candidate nodes in the pool and select the node with the largest acquisition function value as the query node for the oracle.

The major challenge of performance-based methods lies in how to infer the change in the model's parameters or predictions after the addition of the labeled candidate node into the training set without actually re-training the model. Even though we can obtain the exact change of the model parameters or predictions via re-training the model with the newly added labeled candidate node, the computational overhead is unbearable for practical use since the number of unlabeled nodes is quite large. The acquisition function in our proposed method is delicately designed for approximating such change in GNN parameters or node embeddings if the candidate node is added to the labeled node set.

### B.3.2 Expected Model Change Maximization (EMCM) in General Active Learning

As an important subcategory of performance-based methods, the model-change-based idea has been studied in the general AL literature [6, 5]. We first present the general framework of EMCM. After that, we give an empirical interpretation of EMCM to motivate this idea.

The objective of transductive semi-supervised learning can be represented as learning a function $f(\cdot)$ parameterized with $\boldsymbol{\Theta}$, which minimizes the empirical loss $\mathcal{L}$ on the given training data $D = \{D_l, D_u\}$ with the labeled data $\{(\boldsymbol{x}_i, \boldsymbol{y}_i)\}_{i \in D_l}$ and unlabeled data $\{\boldsymbol{x}_i\}_{i \in D_u}$.

$$\mathcal{L} = \sum_{i \in D_l} \ell(f(\boldsymbol{x}_i; \boldsymbol{\Theta}), \boldsymbol{y}_i) + \mathcal{L}' \tag{14}$$

Here, the training loss $\mathcal{L}$ consists of two terms: the supervised loss term $\sum_{i \in D_l} \ell(f(\boldsymbol{x}_i; \boldsymbol{\Theta}), \boldsymbol{y}_i)$ and the unsupervised loss term $\mathcal{L}'$ without the use of label information. $\ell(\cdot, \cdot)$ can be set as the cross

entropy loss in the classification task and squared loss in the regression task. To minimize $\mathcal{L}$ in Eq. (14), we commonly use SGD to search for

$$\boldsymbol{\Theta}^* = \arg\min_{\boldsymbol{\Theta}} \mathcal{L}.$$

In the active learning setting, we consider the influence of a candidate instance $(\boldsymbol{x}_k^+, \boldsymbol{y}_k^+)$ if its label is revealed to the model. Hence, the empirical training loss on the new training set $D_l \cup (\boldsymbol{x}_k^+, \boldsymbol{y}_k^+)$ and $D_u \setminus \boldsymbol{x}_k^+$ becomes as

$$\mathcal{L}^{k,\boldsymbol{y}_k^+} = \mathcal{L} + \ell(f(\boldsymbol{x}_k^+; \boldsymbol{\Theta}), \boldsymbol{y}_k^+). \tag{15}$$

Eq. (15) is called the look-ahead model in the active learning literature. Note that in the transductive setting, the model has access to all samples $\{\boldsymbol{x}_i\}_{i \in D}$ in the training data $D$ and the unsupervised loss $\mathcal{L}'$ in Eq. (14) does not depend on the label information $\{\boldsymbol{y}_i\}_{i \in D_l}$. Therefore, $\mathcal{L}'$ remains the same when we move the candidate instance from the unlabeled set $D_u$ to the labeled set $D_l$. Similarly, the updated trained parameters becomes as

$$\tilde{\boldsymbol{\Theta}}^* = \arg\min_{\boldsymbol{\Theta}} \mathcal{L}^{k,\boldsymbol{y}_k^+}.$$

The idea of model change maximization [7] aims to select the sample $k^*$ that leads to the maximum model change and we can formulate the corresponding acquisition function as

$$k^* = \arg\max_{k \in D_u} \mathcal{A}(k) = \arg\max_{k \in D_u} \| \arg\min_{\boldsymbol{\Theta}} \mathcal{L}^{k,\boldsymbol{y}_k^+} - \arg\min_{\boldsymbol{\Theta}} \mathcal{L} \|_2 = \arg\max_{k \in D_u} \|\tilde{\boldsymbol{\Theta}}^* - \boldsymbol{\Theta}^*\|_2 \quad (16)$$

It is also worth noting that in the original paper [7], we can approximate the model change

$$\|\tilde{\boldsymbol{\Theta}}^* - \boldsymbol{\Theta}^*\|_2 \approx \|\alpha \frac{\partial \mathcal{L}^{k,\boldsymbol{y}_k^+}}{\partial \boldsymbol{\Theta}}\|_2$$

via the SGD update rule $\tilde{\boldsymbol{\Theta}}^* \leftarrow \boldsymbol{\Theta}^* - \alpha \frac{\partial \mathcal{L}^{k,\boldsymbol{y}_k^+}}{\partial \boldsymbol{\Theta}}$ with $\alpha$ as the learning rate. However, the learning rate may not be identical during the training for each candidate sample, we still stick to the exact measure of $\|\tilde{\boldsymbol{\Theta}}^* - \boldsymbol{\Theta}^*\|_2$.

In practical settings, we cannot directly calculate the model change in Eq. (16), since the true label $\boldsymbol{y}_k^+$ of the candidate example $\boldsymbol{x}_k^+$ is unknown before querying. Instead, we extend the idea of model change maximization to the Expected Model Change Maximization (EMCM) [5]. Assume the label lies in the label space $\mathcal{Y}$. For regression tasks, our EMCM criterion for AL is formulated as

$$k^* = \arg\max_{k \in D_u} \mathcal{A}(k) = \arg\max_{k \in D_u} \int_{\mathcal{Y}} \mathbb{P}(\boldsymbol{y}_k^+ \mid \boldsymbol{x}_k^+) \|\tilde{\boldsymbol{\Theta}}^* - \boldsymbol{\Theta}^*\|_2 d\boldsymbol{y}_k^+.$$

For the classification task, our EMCM criterion for AL is formulated as

$$k^* = \arg\max_{k \in D_u} \mathcal{A}(k) = \arg\max_{k \in D_u} \sum_{\boldsymbol{y}_k^+ \in \mathcal{Y}} \mathbb{P}(\boldsymbol{y}_k^+ \mid \boldsymbol{x}_k^+) \|\tilde{\boldsymbol{\Theta}}^* - \boldsymbol{\Theta}^*\|_2.$$

We finally introduce a high-level intuitive interpretation of the EMCM principle in AL. The model change is a reasonable indicator for reducing the generalization error for the following two major reasons. First, The generalization capability can be changed if and only if the current model is changed. As a result, it is useless to query the instance that cannot update the current model in AL. Second, The data points significantly changing the current model are expected to produce a faster convergence speed to the true model, and this is the underlying motivation behind the EMCM framework.

We here note that a big change in the current model does not always lead to better generalization performance, since an outlier also results in a big model change. However, in AL tasks, unlabeled examples are repeatedly selected from a given pool set. Once the model has been changed by an outlier, the EMCM strategy will certainly query a good example in the next data selection iteration that maximizes the change again, which immediately relieves the negative effect of the outlier. In practice, because the amount of outliers is usually very restricted in the data, it is reasonable to believe that the EMCM framework will result in very good generalization performance with more data instances queried.

# C   More Discussions of the Proposed Method

## C.1   The Bayesian Probabilistic Learning Framework for GNNs under SSL

### C.1.1   Active Learning and Semi-supervised Learning

The relationship between active learning and semi-supervised learning lies in their common objective of making the most of limited labeled data. In fact, active learning can be seen as a way to select the most informative instances from an unlabeled dataset to be labeled under semi-supervised learning. By actively querying labels for the most informative instances, active learning can help to effectively utilize the limited labeling resources in a semi-supervised learning setup. Therefore, our starting point is to revisit the training process of GNNs under the semi-supervised setting. Note that in the node selection step in graph active learning in Figure 5, we are actually faced with a semi-supervised node classification task with the current labeled node set and we want to infer what the predictions would be if we add one candidate node into the labeled node set using the look-ahead model. Our recap of the GNNs under the semi-supervised learning setting in Sec. 3.1 provides an alternative view of the model predictions, making it possible to efficiently approximate the expected results without re-training GNNs in the look-ahead model.

### C.1.2   Advantages of Bayesian Interpretation of Backbone GNN Models for Active Learning

Bayesian Interpretations of the backbone GNNs can help to design the acquisition function for active learning in a more interpretable and principled manner. First, many active learning methods are uncertainty-based by selecting the most uncertain node in the pool of the unlabeled nodes since we can assume that the node that the model is most unsure about should reside near the decision boundary. By selecting such uncertain nodes for query, the model may gain the most valuable information about the underlying distribution of the input samples. However, many backbone GNN models like GCN do not possess such Bayesian interpretations in nature so the uncertainty quantification by GNNs directly is not convincing for the design of the subsequent acquisition function. Therefore, we prefer the backbone GNN models that could have clear Bayesian interpretations for the uncertainty-based active learning methods. Second, some performance-based active learning methods by analyzing the influence of a candidate node could have the model performance or model parameters if its label is added to the training set by the oracle. The look-ahead model introduced in Appendix B.3.1 is widely used in performance-based methods and the predictions in the look-ahead model should be efficiently obtained without re-training after adding the new candidate node. By providing the Bayesian interpretations of the backbone GNNs, we actually present an alternative view of the output of the prediction or node embeddings by GNN models from the probabilistic perspective. Therefore, we can now efficiently update the model predictions after observing the new candidate node in the labeled set without re-training. In this work, we resort to a Bayesian interpretation of the GNNs training due to the second reason since our method extends the EMCM method in a general active learning setting, which is also a performance-based method. We leave the extension of other uncertainty-based methods with our proposed Bayesian learning framework for future work as it is out of the scope of this paper.

Some may argue that Bayesian GNNs have well-defined Bayesian interpretations. However, if we use Bayesian GNNs as the backbone model, its training overhead is too huge to be employed in the active learning process in Figure 4. Note that the training step in each AL iteration is inevitable in the active learning setting. Besides, it is extremely difficult to find a good approximation of the predictions in the look-ahead model for performance-based methods like EMCM since Bayesian GNNs also treat the training parameters as distributions instead of point estimates, and treat graph structures as random variables instead of the given fixed input. The resulting computed acquisition function for each candidate node would also be the distributions instead of a real value. Therefore, we propose a light Bayesian learning framework for GNNs and give a clear Bayesian interpretation for GNNs without treating the model parameters as distributions. We leave the investigation of Bayesian GNNs as the backbone of future work since it is also out of the scope of this paper.

### C.1.3   Incorporation of other GNNs

Theorem 3.2 presents an equivalent view of Eq. (5) from the Bayesian perspective. Although it is derived based on the SGC model, other GNNs can also be incorporated into this Bayesian learning

framework Eq. (6) without too many difficulties thanks to the unified optimization framework regarding the forward pass of GNNs [102]. We list some other GNN models here. To make the notations more compact, we omit the training parameters here and focus on the node embeddings $\mathbf{U}$ or $\mathbf{u}$ instead.

**GCN**   Following Theorem 3.2 in [102], we consider the one-layer GCN without the activation function.

**Theorem C.1** (Theorem 3.2 in [102]). *The forward pass of a one-layer GCN,* $\mathbf{U}(\boldsymbol{\Theta}) = \hat{\tilde{\mathbf{A}}}\mathbf{X}\boldsymbol{\Theta}$, *optimizes the following problem :*

$$\mathbf{U}(\boldsymbol{\Theta}) = \arg\min_{\mathbf{U}} \mathrm{Tr}(\mathbf{U}^{\intercal}\tilde{\mathbf{L}}\mathbf{U}) + \|\mathbf{U} - \mathbf{X}\boldsymbol{\Theta}\|_F^2.$$

*Here,* $\tilde{\mathbf{L}} = \mathbf{I} - \hat{\tilde{\mathbf{A}}}$ *is the normalized Laplacian matrix. Then,*

$$\mathbf{U}(\boldsymbol{\Theta}) = \hat{\tilde{\mathbf{A}}}\mathbf{X}\boldsymbol{\Theta}.$$

Then for the binary node classification task, the relaxed single-level optimization problem becomes as

$$\min_{\mathbf{u}} \mathcal{L} = \sum_{i \in \mathcal{V}_l} \ell(\sigma(u_i), y_i) + \mathrm{Tr}\left(\mathbf{u}^{\intercal}\tilde{\mathbf{L}}_\lambda \mathbf{u}\right) + \|\mathbf{u} - \mathbf{X}\boldsymbol{\Theta}\|_2^2 \tag{17}$$

**PPNP/APPNP**   PPNP/APPNP [33] is a graph neural network that utilizes a propagation mechanism derived from personalized PageRank and separates the feature transformation from the aggregation process.

**Theorem C.2** (Theorem 3.3 in [102]). *The forward pass of PPNP/APPNP optimizes the following problem :*

$$\mathbf{U} = \arg\min_{\mathbf{U}} \mathrm{Tr}(\mathbf{U}^{\intercal}\tilde{\mathbf{L}}\mathbf{U}) + \xi\|\mathbf{U} - \mathit{MLP}(\mathbf{X})\|_F^2.$$

*Here,* $\tilde{\mathbf{L}} = \mathbf{I} - \hat{\tilde{\mathbf{A}}}$ *is the normalized Laplacian matrix and* $\mathit{MLP}(\cdot)$ *is an MLP model.*

Then for the binary node classification task, the relaxed single-level optimization problem becomes as

$$\min_{\mathbf{u}} \mathcal{L} = \sum_{i \in \mathcal{V}_l} \ell(\sigma(u_i), y_i) + \mathrm{Tr}\left(\mathbf{u}^{\intercal}\tilde{\mathbf{L}}_\lambda \mathbf{u}\right) + \xi\|\mathbf{u} - \mathtt{MLP}(\mathbf{X})\|_F^2. \tag{18}$$

It is easy to see the optimization framework Eq. (18) is quite general and can also admit the GCN optimization framework in Eq. (17). The optimization interpretation of the forward pass of other GNNs can be found in Table 1 in [102]. It is also interesting to see that the classic label propagation algorithm can be recovered if we set $\xi = 0$ in Eq. (18), where we neglect the node features.

Based on Eq. (18), we can obtain the following theorem similar to Theorem 3.2.

**Theorem C.3.** *Solving Eq. (18) is equivalent to finding the maximum a posteriori (MAP) estimate of the posterior probability distribution with the density* $\mathbb{P}(\mathbf{u} \mid \mathbf{y})$ *as*

$$\mathbb{P}(\mathbf{u} \mid \mathbf{y}) \propto \mu(\mathbf{u}) \exp\left(-\Phi_\ell(\mathbf{u})\right). \tag{19}$$

*Here, the prior* $\mu(\mathbf{u}) \propto \exp\left(-\mathrm{Tr}(\mathbf{u}^{\intercal}\tilde{\mathbf{L}}_\lambda \mathbf{u}) - \xi\|\mathbf{u} - \mathit{MLP}(\mathbf{X})\|_F^2\right)$ *and the likelihood* $\exp\left(-\Phi_\ell(\mathbf{u}(\boldsymbol{\Theta}))\right)$ *is defined by the likelihood potential* $\Phi_\ell(\mathbf{u}) := \sum_{i \in \mathcal{V}_l} \ell(\sigma(u_i), y_i)$. $\mathbf{y} = [y_i]_{i \in \mathcal{V}_l} \in \{0, 1\}^{n_l}$.

It is worth noting that now the prior in Theorem C.3 is no longer a Gaussian distribution as in Theorem 3.2 and we cannot get the closed-form MAP solution as in Theorem 3.3 efficiently. Therefore, we stick to the result derived from the SGC model from the theoretical perspective and verify the generalization of the acquisition function to other GNNs from the empirical perspective. More discussions can be found in Appendix C.2.3. We also leave the investigation of the closed-form MAP estimator of other GNNs as future work since it is out of the scope of this work. The derived acquisition function in Eq. (11) has already achieved great performance when it is tested with other GNNs in the experiments (Sec. 4.2 with the paragraph title **Generalization to Other GNNs**), it is fine to skip the more detailed investigation of the corresponding acquisition functions derived from other GNNs.

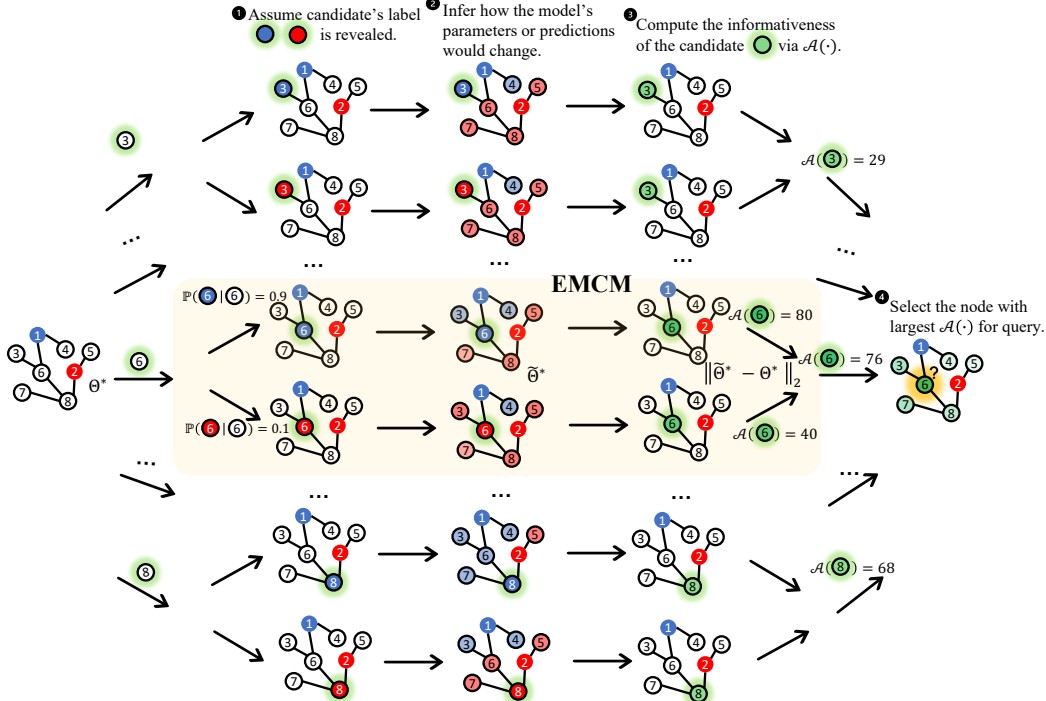

Figure 6: Expected Model Change Maximization (EMCM) for Active Learning on Graphs. The final acquisition function value for each candidate node is the expected model change $\|\widetilde{\boldsymbol{\Theta}}^* - \boldsymbol{\Theta}\|_2$ taken over the prediction probability of the candidate label.

### C.1.4  Clarifications on the Size of Node Embeddings

In the literature, we sometimes may interpret the node embeddings as the output by the intermediate layer of a GNN model and the dimension of the node embedding vector could be arbitrary. But, we have to add an extra linear layer on top of such node embeddings to align the size of the output with the number of the classes before the softmax layer for final predictions. However, in the original papers like GCN [32] and SGC [68], the output dimension of the node embeddings is termed as the output matrix right before the softmax layer, which exactly $n \times c$, with the number of classes as $c$. We stick to this notation in this paper for analysis.

### C.1.5  Relationships with other Optimization Frameworks Unifying GNNs

From the discussion of Appendix C.1.3, our Bayesian learning framework for GNNs is based on the existing optimization framework unifying the forward pass of GNNs [102]. Our Bayesian learning framework extends the unified optimization framework and incorporates the backward pass of GNN training for the semi-supervised node classification task as well, bringing new insights into the interpretations of GNN models from the probabilistic aspect. There are also several recent works that unify GNNs as well. ADA-UGNN [42] unifies GNNs from the perspective of graph signal denoising and it also neglects the backward pass of GNN training as [102]. A very recent framework [23] indeed considers both the forward and backward passes of GNNs and can be viewed as another extension of Eq. (18) by viewing both the node embeddings and the model parameters as optimization variables. This framework may admit more existing GNNs but it hinders the incorporation of Bayesian interpretations and further employment in the look-ahead model for active learning. Our proposed Bayesian learning framework for GNNs is lightweight and specifically designed for efficient integration in the look-ahead model for active learning.

## C.2 Expected Model Change Maximization for Graph Active Learning

### C.2.1 Illustrations of EMCM for AL on Graphs

Eq. (7) extends the general EMCM principle to the GNN model, as illustrated in Figure 6. Based on the discussions of EMCM in general AL (appendix B.3), we know that in practice, due to the unavailability of the label for the candidate node during node selection, we can use the prediction probability of the candidate node by the current model to obtain the expected model changes in terms of model parameters in the EMCM acquisition function. For example, in Figure 6, the current model makes the following predictions on candidate node 6, with the probability of 0.9 to be blue and the probability of 0.1 to be red. Then by the look-ahead model, if we query node 6 for its label and add it to the labeled node set, the model parameters will be updated accordingly from $\boldsymbol{\Theta}^*$ to $\tilde{\boldsymbol{\Theta}}^*$. We then calculate the change of parameters as $\|\tilde{\boldsymbol{\Theta}}^* - \boldsymbol{\Theta}^*\|_2$ under all cases of potential labels. The final acquisition function is to take the expectation of the model change with respect to the predicted class probability. In this example, the final acquisition function value is computed as $80 \times 0.9 + 40 \times 0.1 = 76$.

Eq. (7) actually measures the change of node embeddings directly as $\|\mathbf{u}(\tilde{\boldsymbol{\Theta}}^*) - \mathbf{u}(\boldsymbol{\Theta}^*)\|_2$, instead of the model parameters as $\|\tilde{\boldsymbol{\Theta}}^* - \boldsymbol{\Theta}^*\|_2$. We make this change to obtain a more elegant closed-form solution for efficiency since the proposed Bayesian learning framework focuses on the distribution of the node embeddings instead of the model parameters. Moreover, the change in node embeddings is equivalent to the change in model parameters in the SGC model since we fix $\mathbf{u}(\boldsymbol{\Theta}) = \mathbf{X}\boldsymbol{\Theta}$. The only assumption is the orthogonality of the node feature matrix $\mathbf{X}^\mathsf{T}\mathbf{X} = \mathbf{I}$ and it generally holds when we use the node id (basis vector) for the node feature vector. Since the $l_2$ norm is orthogonality invariant, we know that $\|\mathbf{u}(\tilde{\boldsymbol{\Theta}}^*) - \mathbf{u}(\boldsymbol{\Theta}^*)\|_2 = \|\mathbf{X}\tilde{\boldsymbol{\Theta}}^* - \mathbf{X}\boldsymbol{\Theta}^*\|_2 = \|\tilde{\boldsymbol{\Theta}}^* - \boldsymbol{\Theta}^*\|_2$ when we use the SGC model as the backbone. We leave the more general case without such an orthogonality assumption on the node feature matrix $\mathbf{X}$ as future work and it is out of the scope of this paper by now.

### C.2.2 Bayesian Learning Framework via Truncated Spectral Projection

We add more details of Eq. (8) in the look-ahead model. Note that from Eq. (5), we have $\boldsymbol{\alpha} = \mathbf{V}^\mathsf{T}\mathbf{u}(\boldsymbol{\Theta})$ and thus $\mathbf{V}\boldsymbol{\alpha} = \mathbf{u}(\boldsymbol{\Theta})$.

$$
\min_{\substack{\boldsymbol{\Theta} \\ \mathbf{u}(\boldsymbol{\Theta})=\mathbf{X}\boldsymbol{\Theta}}} \mathcal{L} = \sum_{i \in \mathcal{V}_l} \ell(\sigma(u_i(\boldsymbol{\Theta})), y_i) + \mathrm{Tr}\left(\mathbf{u}(\boldsymbol{\Theta})^\mathsf{T}\tilde{\mathbf{L}}_\lambda \mathbf{u}(\boldsymbol{\Theta})\right)
$$
$$
\Leftrightarrow \min_{\mathbf{V}\boldsymbol{\alpha}} \mathcal{L} = \sum_{i \in \mathcal{V}_l} \ell(\sigma(\mathbf{e}_i^\mathsf{T}\mathbf{V}\boldsymbol{\alpha}), y_i) + \mathrm{Tr}\left(\boldsymbol{\alpha}^\mathsf{T}\mathbf{V}^\mathsf{T}\tilde{\mathbf{L}}_\lambda \mathbf{V}\boldsymbol{\alpha}\right)
$$
$$
\Leftrightarrow \min_{\boldsymbol{\alpha}} \tilde{\mathcal{L}} = \sum_{i \in \mathcal{V}_l} \ell(\sigma(\mathbf{e}_i^\mathsf{T}\mathbf{V}\boldsymbol{\alpha}), y_i) + \mathrm{Tr}\left(\boldsymbol{\alpha}^\mathsf{T}\boldsymbol{\Lambda}_\lambda \boldsymbol{\alpha}\right)
$$

The above derivation holds since $\mathbf{V}^\mathsf{T}\mathbf{V} = \mathbf{I}$.

### C.2.3 Potential Use of other GNNs

Although the acquisition function in Eq. (11) is specifically designed for the SGC model, we can still use other GNN models as the graph representation model $\mathcal{M}$ in Algorithm 2 without anything else changed. The resulting method still works because other GNNs only lead to more complex forms of the optimization framework in Eq. (5) and the corresponding MAP estimator in Eq. (6), but they are all the same in nature from the Bayesian perspective. The forward pass of any GNN model defines a prior (not necessarily Gaussian now) over the node embeddings while the corresponding backward pass specifies the likelihood based on the supervision loss function, working together to determine the posterior. Therefore, Eq. (11) can be viewed as an approximation of the exact value of model change if the backbone model is any other GNN model. The derived acquisition function in Eq. (11) has already achieved great performance when it is tested with other GNNs in the experiments (Sec. 4.2 with the paragraph title **Generalization to Other GNNs**). We leave the investigation of the closed-form MAP estimator of other GNNs as future work since it is out of the scope of this work.

### C.2.4 Multi-class Classification Extension with Cross-Entropy Loss

In this section, we derive all the necessary formulas when we apply EMCM method on graph active learning for multi-class classification with the cross-entropy loss. Most of the results are adapted from [47]. Recall that, the cross-entropy loss function

$$\ell(\boldsymbol{x}, \boldsymbol{y}) = -\sum_{i=1}^{c} \boldsymbol{x}_i \ln(\boldsymbol{y}_i),$$

and we want to incorporate it into the multi-class optimization framework Eq. (4) first. The cross-entropy loss requires that both inputs are probability distribution vectors on the set of possible classes, $\{1, \cdots, c\}$. While the observations $\boldsymbol{y}_i$ satisfy this property because of their one-hot form, the rows of arbitrary $\mathbf{U}(\boldsymbol{\Theta}) \in \mathbb{R}^{n \times c}$ do not necessarily satisfy this same condition. The entries of $\mathbf{U}(\boldsymbol{\Theta})$ are not even constrained to be non-negative. As such, following the common practice in the field, we apply the Softmax function $\mathsf{S}(\cdot)$ on the rows of $\mathbf{U}(\boldsymbol{\Theta})$ to enforce this probability distribution property. Denoting the (i,c)-th entry of $\mathbf{U}(\boldsymbol{\Theta})$ by $\mathbf{U}_{i,c}(\boldsymbol{\Theta})$ and the $c$-th entry of $\boldsymbol{y}_i$ by $[\boldsymbol{y}_i]_c$, we have

$$\mathsf{S}(\mathbf{U}_i(\boldsymbol{\Theta})) \coloneqq \frac{1}{\sum_{j=1}^{c} \exp(\mathbf{U}_{i,j}(\boldsymbol{\Theta}))} \left( \exp\left(\mathbf{U}_{i,1}\right), \cdots, \exp\left(\mathbf{U}_{i,c}\right) \right)^{\mathsf{T}}.$$

Therefore, we can rewrite Eq. (4) as

$$\min_{\substack{\boldsymbol{\Theta} \\ \mathbf{U}(\boldsymbol{\Theta}) = \mathbf{X}\boldsymbol{\Theta}}} \mathcal{L} = \sum_{i \in \mathcal{V}_l} \left\{ -\boldsymbol{y}_i^{\mathsf{T}} \mathbf{U}(\boldsymbol{\Theta}) + \ln\left( \sum_{j=1}^{c} \exp\left(\mathbf{U}_{i,j}(\boldsymbol{\Theta})\right) \right) \right\} + \mathrm{Tr}\left( \mathbf{U}(\boldsymbol{\Theta})^{\mathsf{T}} \tilde{\mathbf{L}}_{\lambda} \mathbf{U}(\boldsymbol{\Theta}) \right). \quad (20)$$

Next, we apply Laplacian approximation and truncated spectral projection for the cross-entropy model. For ease in calculations, we let the vector $\mathbf{u}(\boldsymbol{\Theta}) \in \mathbb{R}^{nc}$ be the concatenation of the columns of $\mathbf{U}(\boldsymbol{\Theta})$. Likewise, define $\boldsymbol{\alpha} \in \mathbb{R}^{mc}$ to be the concatenation of the columns of the matrix $\mathbf{A}$ ($\mathbf{A} = \mathbf{V}^{\mathsf{T}} \mathbf{U}(\boldsymbol{\Theta})$). Define

$$\mathcal{V} \coloneqq \mathrm{diag}(V, V, \ldots, V) \in \mathbb{R}^{nc \times mc},$$

and

$$\boldsymbol{\Lambda}_{\lambda}^{\otimes} \coloneqq \mathrm{diag}\left(\boldsymbol{\Lambda}_{\lambda}, \boldsymbol{\Lambda}_{\lambda}, \ldots, \boldsymbol{\Lambda}_{\lambda}\right) \in \mathbb{R}^{mc \times mc}.$$

Define $\mathcal{P}_i \in \mathbb{R}^{c \times nc}$ to be the projection matrix that picks out the indices in $\mathbf{u}(\boldsymbol{\Theta}) \in \mathbf{R}^{nc}$ corresponding to node $i$; i.e., selecting the $i$-th row of the matrix $\mathbf{U}(\boldsymbol{\Theta}) \in \mathbb{R}^{n \times c}$. Then $\mathbf{u}(\boldsymbol{\Theta}) = \mathcal{V}\boldsymbol{\alpha}$, and with $\hat{\mathbf{e}}_j$ denoting the $j$-th standard basis vector in $\mathbb{R}^{nc}$, the truncated spectral projection for cross-entropy model's objective (similar to Eq. (8)) can be written as

$$\min_{\boldsymbol{\alpha}} \tilde{\mathcal{L}} = \sum_{i \in \mathcal{V}_l} \left\{ -\boldsymbol{y}_i^{\mathsf{T}} \mathcal{P}_i \mathcal{V} \boldsymbol{\alpha} + \ln\left( \sum_{j=1}^{c} \exp\left(\boldsymbol{\alpha}^{\mathsf{T}} \mathcal{V}^{\mathsf{T}} \hat{\mathbf{e}}_{i+(j-1)n}\right) \right) \right\} + \mathrm{Tr}\left( \boldsymbol{\alpha}^{\mathsf{T}} \boldsymbol{\Lambda}_{\lambda}^{\otimes} \boldsymbol{\alpha} \right). \quad (21)$$

Define

$$\pi_c^i \coloneqq \frac{\exp\left(\boldsymbol{\alpha}^{\mathsf{T}} \mathcal{V}^{\mathsf{T}} \hat{\mathbf{e}}_{i+(c-1)n}\right)}{\sum_{j=1}^{c} \exp\left(\boldsymbol{\alpha}^{\mathsf{T}} \mathcal{V}^{T} \hat{\mathbf{e}}_{i+(j-1)n}\right)} \quad \text{and} \quad \pi^i \coloneqq \left(\pi_1^i, \cdots, \pi_c^i\right)^T \in \mathbb{R}^c.$$

The Laplace approximation for the CE model yields

$$\boldsymbol{\alpha} \mid \mathbf{Y} \sim \mathcal{N}(\hat{\boldsymbol{\alpha}}, \hat{\mathbf{C}}_{\hat{\boldsymbol{\alpha}}}), \hat{\boldsymbol{\alpha}} = \arg\min_{\boldsymbol{\alpha}} \tilde{\mathcal{L}}, \hat{\mathbf{C}}_{\hat{\boldsymbol{\alpha}}} = \left( \boldsymbol{\Lambda}_{\lambda}^{\otimes} + \mathcal{V}^{T} \left( \mathbf{D}_{\mathcal{L}}(\boldsymbol{\alpha}) - \Pi_{\mathcal{L}}(\boldsymbol{\alpha}) \Pi_{\mathcal{L}}^{T}(\boldsymbol{\alpha}) \right) \mathcal{V} \right)^{-1}. \quad (22)$$

Here,

$$D_{\mathcal{L}}\left(\pi_c(\boldsymbol{\alpha})\right) = \sum_{i \in \mathcal{L}} \pi_c^i(\boldsymbol{\alpha}) \mathbf{e}_i \mathbf{e}_i^T, \quad \Pi_{\mathcal{L}}(\boldsymbol{\alpha}) = \begin{pmatrix} D_{\mathcal{L}}\left(\pi_1(\boldsymbol{\alpha})\right) \\ D_{\mathcal{L}}\left(\pi_2(\boldsymbol{\alpha})\right) \\ \vdots \\ D_{\mathcal{L}}\left(\pi_c(\boldsymbol{\alpha})\right) \end{pmatrix},$$

$$\mathbf{D}_{\mathcal{L}}(\boldsymbol{\alpha}) = \begin{pmatrix} D_{\mathcal{L}}\left(\pi_1(\boldsymbol{\alpha})\right) & 0 & \cdots & 0 \\ 0 & D_{\mathcal{L}}\left(\pi_2(\boldsymbol{\alpha})\right) & \cdots & \vdots \\ \vdots & \vdots & \ddots & \vdots \\ 0 & \cdots & \cdots & D_{\mathcal{L}}\left(\pi_c(\boldsymbol{\alpha})\right) \end{pmatrix}.$$

The inverse of $\hat{\mathbf{C}}_{\hat{\boldsymbol{\alpha}}} \in \mathbb{R}^{mc \times mc}$ is not prohibitively costly to compute because of its restricted size. The look-ahead MAP estimator is then given as,

$$\hat{\boldsymbol{\alpha}}^{k,\boldsymbol{y}_k^+} = \hat{\boldsymbol{\alpha}} - \mathcal{C}_{\hat{\boldsymbol{\alpha}}} \mathcal{V}_k^T \left( I - T_k^T \left( I + T_k G_k T_k^T \right)^{-1} T_k G_k \right) \left( \pi^k - \boldsymbol{y}_k^+ \right).$$

Here, $G_k = \mathcal{V}_k \mathcal{C}_{\hat{\boldsymbol{\alpha}}} \mathcal{V}_k^T$, $\mathcal{V}_k := \mathcal{P}_k \mathcal{V}$, $B_k := \text{diag} \left( \pi^k \right) - \pi^k \left( \pi^k \right)^T$ and $B_k = T_k^T T_k$. The final acquisition function is given as

$$\mathcal{A}(k) = \sum_{i=1}^{c} \mathbb{P}(\boldsymbol{y}_k^+ = \mathbf{e}_i | k) \left\| \mathcal{C}_{\hat{\boldsymbol{\alpha}}} \mathcal{V}_k^T \left( I - T_k^T \left( I + T_k G_k T_k^T \right)^{-1} T_k G_k \right) \left( \pi^k - \boldsymbol{y}_k^+ \right) \right\|_2. \tag{23}$$

This is efficient to compute because calculating Eq. (23) involves only $c \times c$ matrices.

## D    Pseudocodes

### D.1    Pool-based Active Learning on Graphs

We introduce the popular pool-based active learning setting for graph-structured data [88], where a pool of unlabeled nodes is available for selection or query. Algorithm 1 provides a sketch of the generic pool-based AL for any graph learning model $\mathcal{M}$.

Without the loss of generality, we consider a batch setting with $B/b$ rounds where $b$ nodes are selected in each iteration. The target model is initialized and retrained on all of the labeled data from the previous rounds to avoid any correlation between the selection. In each iteration, we train the model with label supervision from the current labeled node set (line 4). Next, an active learner selects the most valuable $b$ nodes based on the acquisition function (lines 5-9). These $b$ nodes constitute the query node set, and their labels are annotated by the oracle. Finally, we update the labeled and unlabeled node sets accordingly. The ultimate predictions are given by the model after it is trained on the augmented labeled node set for the last time (lines 13-21). The core of active learning is to design the acquisition function $\mathcal{A}(\cdot)$.

---

**Algorithm 1:** Pool-based Active Learning Setting on Graphs

**Input**  : Graph $\mathcal{G} = (\mathcal{V}, \mathcal{E})$ with $\mathbf{A}, \mathbf{X}$, initial labeled node set $\mathcal{V}_l$, query batch size $b$, labeling budget $B$, acquisition function $\mathcal{A}(\cdot)$.
**Output**: Predictions $\hat{y}_i$ for all nodes $i \in \mathcal{V}$.

1  $\mathcal{V}_l^{(0)} \leftarrow \mathcal{V}_l, \mathcal{V}_u^{(0)} \leftarrow \mathcal{V} \setminus \mathcal{V}_l$
2  /* Augment the labeled node set with the oracle labeling budget $B$.  */
3  **for** $t \leftarrow 0$ **to** $B/b - 1$ **do**
4  $\quad$ Train a model $\mathcal{M}$ based on $\mathcal{G}$ and $\mathcal{V}_l^{(t)}$  // Train the model with current labeled set.
5  $\quad$ $\mathcal{V}_q = \phi$                                    // Initialize the query node set as empty.
6  $\quad$ **for** $k = 1, 2, \cdots, b$ **do**
7  $\quad\quad$ $k^* = \arg\max_{k \in \mathcal{V}_u^{(t)}} \mathcal{A}(k)$    // Select $b$ most informative nodes based on $\mathcal{A}(\cdot)$.
8  $\quad\quad$ $\mathcal{V}_q \leftarrow \mathcal{V}_q \cup \{k^*\}$                      // Query the oracle for labels.
9  $\quad$ **end**
10 $\quad$ $\mathcal{V}_l^{(t+1)} \leftarrow \mathcal{V}_l^{(t)} \cup \mathcal{V}_q, \mathcal{V}_u^{(t+1)} \leftarrow \mathcal{V}_u^{(t)} \setminus \mathcal{V}_q$ // Update labeled and unlabeled node sets.
11 **end**
12 /* Return the predictions for all nodes.  */
13 Train the model $\mathcal{M}$ based on $\mathcal{V}_l^{\left(\frac{B}{b}\right)}$      // Train the model with the latest labeled set.
14 **for** $i \in \mathcal{V}$ **do**
15 $\quad$ **if** $i \in \mathcal{V}_l^{\left(\frac{B}{b}\right)}$ **then**
16 $\quad\quad$ $\hat{y}_i \leftarrow y_i$               // Return the ground-truth if the node has been labeled.
17 $\quad$ **end**
18 $\quad$ **else**
19 $\quad\quad$ $\hat{y}_i$ is set based on the predictions of the latest trained model $\mathcal{M}$
20 $\quad$ **end**
21 **end**
22 **return** $\{\hat{y}_i\}_{i \in \mathcal{V}}$

---

## D.2 DOCTOR Algorithm for Binary Classification under Sequential Active Learning Setting

We first present the basic version of the DOCTOR algorithm with the sequential active learning setting for binary classification (Algorithm 2). In the sequential active learning setting, we only select the most informative node for labeling by the oracle one by one, which means that the query batch size $b = 1$.

Algorithm 2 strictly follows the procedure in Algorithm 1 with $b = 1$ and the training model $\mathcal{M}$ as the SGC. The key step is lines 7-10, where we select the node that leads to the maximum expected model change as the query node based on Eq. (11).

---

**Algorithm 2:** Expected Model Change Maximization on Graphs (DOCTOR) under the Sequential Active Learning Setting for Binary Node Classification

---

**Input** : Graph $\mathcal{G} = (\mathcal{V}, \mathcal{E})$ with $\mathbf{A}, \mathbf{X}$, initial labeled node set $\mathcal{V}_l$, labeling budget $B$.
**Output**: Predictions $\hat{y}_i$ for all nodes $i \in \mathcal{V}$.

1   $\mathcal{V}_l^{(0)} \leftarrow \mathcal{V}_l, \mathcal{V}_u^{(0)} \leftarrow \mathcal{V} \setminus \mathcal{V}_l$

2   Compute $\mathbf{V}$ based on $\tilde{\mathbf{L}}_\lambda^\intercal$ for truncated spectral projection

3   /* Sequential active learning setting. */

4   **for** $t \leftarrow 0$ **to** $B - 1$ **do**

5      Train an SGC model to obtain the node embedding $\mathbf{u}(\boldsymbol{\Theta}_t)$ based on $\mathcal{G}$ and $\mathcal{V}_l^{(t)}$ (Eq. (5) or Eq. (4))

6      Obtain MAP estimator with mean $\hat{\boldsymbol{\alpha}} = \mathbf{V}^\intercal \mathbf{u}(\boldsymbol{\Theta}_t)$ and covariance $\hat{\mathbf{C}}_{\hat{\boldsymbol{\alpha}}}$ (Eq. (9))

7      **for** $k \in \mathcal{V}_u^{(t)}$ **do**

8          Record the value of $\mathcal{A}(k)$ (Eq. (11))    // Expected model change w/o re-training.

9      **end**

10      Select $k^* = \arg\max_{k \in \mathcal{V}_u^{(t)}} \mathcal{A}(k)$             // Select the query node.

11      $\mathcal{V}_l^{(t+1)} \leftarrow \mathcal{V}_l^{(t)} \cup \{k^*\}, \mathcal{V}_u^{(t+1)} \leftarrow \mathcal{V}_u^{(t)} \setminus \{k^*\}$   // Update labeled and unlabeled sets.

12   **end**

13   /* Return the predictions for all nodes. */

14   Train the SGC model based on $\mathcal{V}_l^{(B)}$

15   **for** $i \in \mathcal{V}$ **do**

16      **if** $i \in \mathcal{V}_l^{(B)}$ **then**

17          $\hat{y}_i \leftarrow y_i$        // Return the ground-truth if the node has been labeled.

18      **end**

19      **else**

20          $\hat{y}_i$ is set based on the predictions of the latest trained SGC model.

21      **end**

22   **end**

23   **return** $\{\hat{y}_i\}_{i \in \mathcal{V}}$

---

## D.3 DOCTOR Algorithm for Binary Classification under Batch Active Learning Setting

In the batch active learning, the size of the query node set is larger than 1 ($b > 1$). Therefore, there is an added difficulty of how to choose this subset in the optimal way. Sub-optimal results could not be avoided if we directly choose the top $b$ maximizers of the current values of $\{\mathcal{A}(k)\}_{k \in \mathcal{V}_u^{(t)}}$, as these maximizers often are close in the underlying embedding space in a sense wasting the precious query budget on redundant information.

Following the seminal work [17], we restrict the set of node indices on which the acquisition function $\mathcal{A}(\cdot)$ can evaluate from the whole currently unlabeled set to its smaller subset $\mathcal{S} \subset \mathcal{V}_u$. In other words, we now select the batch query set $\mathcal{V}_q \subset \mathcal{S}$ to be the top $b$ maximizers of the designed acquisition function, where $\mathcal{S}$ is chosen uniformly at random from $\mathcal{V}_u$.

This has essentially two important and positive consequences. First, evaluating $\mathcal{A}(\cdot)$ only on $\mathcal{S}$ is obviously computationally faster since $|\mathcal{S}| \ll |\mathcal{V}_u|$. Second, by selecting $\mathcal{S} \subset \mathcal{V}_u$ at random, we partially alleviate the problem of redundant calculations since the maximizers of $\mathcal{A}(\cdot)$ over $\mathcal{S}$ likely do not lie all close together. We apply this query set selection method to our batch active learning

experiments. The corresponding algorithm is presented in Algorithm 3, a batch query version of Algorithm 2. The key changes are as follows. Now we generate $\mathcal{S} \subset \mathcal{V}_u^{(t)}$ uniformly at random (line 7) and then select the top $b$ maximizers of the acquisition function (lines 8-11).

Extending sequential active learning methods to the batch active learning setting can pose certain challenges. Sequential active learning methods often exploit dependencies between labeled and unlabeled instances. In the batch setting, these dependencies can become more complex and interwoven, especially when considering uncertainty or diversity measures across multiple instances simultaneously. Designing strategies to capture and leverage these dependencies can be non-trivial. Determining the appropriate batch size can be challenging. A larger batch size may increase efficiency but can also introduce more uncertainty or diversity among instances, making it harder to make confident labeling decisions. Conversely, a smaller batch size may not fully utilize the benefits of the batch setting. In sequential AL, diversity is naturally preserved as new instances are added one at a time. However, in the batch setting, there is a risk of losing diversity when selecting instances collectively. Ensuring that the selected batch covers a diverse range of regions in the feature space becomes challenging, and specialized sampling strategies are necessary.

In this work, we focus on the design of the acquisition function in the sequential active learning setting and extend the proposed method in sequential AL to the batch AL with fixed query batch size via the random sampling of the unlabeled pool. While an interesting question, this work does not explore varying the batch size nor the candidate set size, we leave a more in-depth investigation of the batch AL setting for future work as it is currently out of the scope of this paper.

---

**Algorithm 3:** Expected Model Change Maximization on Graphs (**DOCTOR**) under the Batch Active Learning Setting Binary Node Classification

---

**Input** : Graph $\mathcal{G} = (\mathcal{V}, \mathcal{E})$ with $\mathbf{A}, \mathbf{X}$, initial labeled node set $\mathcal{V}_l$, query batch size $b$, labeling budget $B$.
**Output :** Predictions $\hat{y}_i$ for all nodes $i \in \mathcal{V}$.

1   $\mathcal{V}_l^{(0)} \leftarrow \mathcal{V}_l, \mathcal{V}_u^{(0)} \leftarrow \mathcal{V} \setminus \mathcal{V}_l$
2   Compute $\mathbf{V}$ based on $\tilde{\mathbf{L}}_\lambda^\intercal$ for truncated spectral projection
3   /* Batch active learning setting. */
4   **for** $t \leftarrow 0$ **to** $B/b - 1$ **do**
5      Train an SGC model to obtain the node embedding $\mathbf{u}(\mathbf{\Theta}_t)$ based on $\mathcal{G}$ and $\mathcal{V}_l^{(t)}$
6      Obtain MAP estimator with mean $\hat{\boldsymbol{\alpha}} = \mathbf{V}^\intercal \mathbf{u}(\mathbf{\Theta}_t)$ and covariance $\hat{\mathbf{C}}_{\hat{\boldsymbol{\alpha}}}$ (Eq. (9))
7      Generate $\mathcal{S} \subset \mathcal{V}_u^{(t)}$ uniformly at random      // Generate $\mathcal{S}$ with $|\mathcal{S}| \ll |\mathcal{V}_u|$.
8      **for** $k \in \mathcal{S}$ **do**
9         Record the value of $\mathcal{A}(k)$ (Eq. (11))    // Expected model change w/o re-training.
10      **end**
11      Select top $b$ maximizers of $\{\mathcal{A}(k)\}_{k \in \mathcal{S}}$ as $\mathcal{V}_q$      // Select the query node set.
12      $\mathcal{V}_l^{(t+1)} \leftarrow \mathcal{V}_l^{(t)} \cup \mathcal{V}_q, \mathcal{V}_u^{(t+1)} \leftarrow \mathcal{V}_u^{(t)} \setminus \mathcal{V}_q$     // Update labeled and unlabeled sets.
13   **end**
14   /* Return the predictions for all nodes. */
15   Train the SGC model based on $\mathcal{V}_l^{(\frac{B}{b})}$
16   **for** $i \in \mathcal{V}$ **do**
17      **if** $i \in \mathcal{V}_l^{(\frac{B}{b})}$ **then**
18         $\hat{y}_i \leftarrow y_i$      // Return the ground-truth if the node has been labeled.
19      **end**
20      **else**
21         $\hat{y}_i$ is set based on the predictions of the latest trained SGC model.
22      **end**
23   **end**
24   **return** $\{\hat{y}_i\}_{i \in \mathcal{V}}$

---

### D.4 DOCTOR Algorithm for Multi-class Classification under Batch Active Learning Setting

We finally present the version of the proposed DOCTOR algorithm (Algorithm 4) for multi-class classification under the batch active learning setting. Algorithm 4 is almost identical to Algorithm 3, except we now use different formulas to compute the acquisition function.

---

**Algorithm 4:** Expected Model Change Maximization on Graphs (**DOCTOR**) under the Batch Active Learning Setting Multi-class Node Classification

---

**Input** : Graph $\mathcal{G} = (\mathcal{V}, \mathcal{E})$ with $\mathbf{A}, \mathbf{X}$, initial labeled node set $\mathcal{V}_l$, query batch size $b$, labeling budget $B$.

**Output :** Predictions $\hat{\boldsymbol{y}}_i$ for all nodes $i \in \mathcal{V}$.

1 $\mathcal{V}_l^{(0)} \leftarrow \mathcal{V}_l, \mathcal{V}_u^{(0)} \leftarrow \mathcal{V} \setminus \mathcal{V}_l$

2 Compute $\mathbf{V}$ based on $\tilde{\mathbf{L}}_\lambda^\intercal$ for truncated spectral projection

3 /* Batch active learning setting. */

4 **for** $t \leftarrow 0$ **to** $B/b - 1$ **do**

5      Train an SGC model to obtain the node embedding $\mathbf{U}(\boldsymbol{\Theta}_t)$ based on $\mathcal{G}$ and $\mathcal{V}_l^{(t)}$

6      Obtain MAP estimator with mean $\hat{\boldsymbol{\alpha}}$ and covariance $\hat{\mathbf{C}}_{\hat{\boldsymbol{\alpha}}}$ (Eq. (22))

7      Generate $\mathcal{S} \subset \mathcal{V}_u^{(t)}$ uniformly at random          // Generate $\mathcal{S}$ with $|\mathcal{S}| \ll |\mathcal{V}_u|$.

8      **for** $k \in \mathcal{S}$ **do**

9          Record the value of $\mathcal{A}(k)$ (Eq. (23))    // Expected model change w/o re-training.

10      **end**

11      Select top $b$ maximizers of $\{\mathcal{A}(k)\}_{k \in \mathcal{S}}$ as $\mathcal{V}_q$          // Select the query node set.

12      $\mathcal{V}_l^{(t+1)} \leftarrow \mathcal{V}_l^{(t)} \cup \mathcal{V}_q, \mathcal{V}_u^{(t+1)} \leftarrow \mathcal{V}_u^{(t)} \setminus \mathcal{V}_q$     // Update labeled and unlabeled sets.

13 **end**

14 /* Return the predictions for all nodes. */

15 Train the SGC model based on $\mathcal{V}_l^{(\frac{B}{b})}$

16 **for** $i \in \mathcal{V}$ **do**

17      **if** $i \in \mathcal{V}_l^{(\frac{B}{b})}$ **then**

18          $\hat{\boldsymbol{y}}_i \leftarrow y_i$          // Return the ground-truth if the node has been labeled.

19      **end**

20      **else**

21          $\hat{\boldsymbol{y}}_i$ is set based on the predictions of the latest trained SGC model.

22      **end**

23 **end**

24 **return** $\{\hat{y}_i\}_{i \in \mathcal{V}}$

---

## E  Proofs

We present the detailed proofs in this section and some of them are directly adapted from [47].

### E.1  Theorem 3.1

*Proof.* We will prove Theorem 3.1 by induction. The optimization problem is as follows.

$$\min_{\hat{\boldsymbol{\Theta}}} O(\hat{\boldsymbol{\Theta}}) = \mathrm{Tr}(\mathbf{U}^\intercal \tilde{\mathbf{L}} \mathbf{U}), \quad \text{s.t.} \quad \mathbf{U} = \mathbf{X}\hat{\boldsymbol{\Theta}}. \tag{24}$$

Note that we will apply the gradient descent algorithm with the initialization of $\hat{\boldsymbol{\Theta}}$ as $\hat{\boldsymbol{\Theta}}^{(0)} = \boldsymbol{\Theta}$. $\boldsymbol{\Theta}$ can be viewed as a constant in Problem (24) since $\boldsymbol{\Theta}$ is the trainable parameter for SGC model during the feed-forward propagation and it will only be updated during the back-propagation. We are only interested in the forward pass in Theorem 3.1 for now and $\hat{\boldsymbol{\Theta}}$ is the optimization variable in Problem (24) instead of $\boldsymbol{\Theta}$.

*The Basis Step.* When $K = 1$, we start from the initialization $\hat{\boldsymbol{\Theta}}^{(0)} = \boldsymbol{\Theta}$ and $\mathbf{U}^{(0)} = \mathbf{X}\hat{\boldsymbol{\Theta}}^{(0)} = \mathbf{X}\boldsymbol{\Theta}$. Note that we have,

$$\frac{\partial O}{\partial \hat{\boldsymbol{\Theta}}} = \frac{\partial O}{\partial \mathbf{U}} \cdot \frac{\partial \mathbf{U}}{\partial \hat{\boldsymbol{\Theta}}} = 2\mathbf{X}^\intercal \tilde{\mathbf{L}} \mathbf{U}.$$

Therefore, when we apply the gradient descent once to update $\hat{\boldsymbol{\Theta}}$ with the step size $\alpha = \frac{1}{2}$, we have,

$$\hat{\boldsymbol{\Theta}}^* = \hat{\boldsymbol{\Theta}}^{(1)} = \hat{\boldsymbol{\Theta}}^{(0)} - \alpha \cdot 2\mathbf{X}^{\mathsf{T}}\tilde{\mathbf{L}}\mathbf{U}^{(0)} = \boldsymbol{\Theta} - \mathbf{X}^{\mathsf{T}}\tilde{\mathbf{L}}\mathbf{X}\boldsymbol{\Theta} = (\mathbf{I} - \mathbf{X}^{\mathsf{T}}\tilde{\mathbf{L}}\mathbf{X})\boldsymbol{\Theta}.$$

The node embedding matrix $\mathbf{U}$ is updated accordingly as

$$\mathbf{U}^{(1)} = \mathbf{X}\hat{\boldsymbol{\Theta}}^{(1)} = (\mathbf{X} - \mathbf{X}\mathbf{X}^{\mathsf{T}}\tilde{\mathbf{L}}\mathbf{X})\boldsymbol{\Theta} = (\mathbf{I} - \mathbf{X}\mathbf{X}^{\mathsf{T}}\tilde{\mathbf{L}})\mathbf{X}\boldsymbol{\Theta}.$$

If we assume the node feature matrix is orthogonal ($\mathbf{X}\mathbf{X}^{\mathsf{T}} = \mathbf{I}$), we immediately have,

$$\mathbf{U}^{(1)} = (\mathbf{I} - \tilde{\mathbf{L}})\mathbf{X}\boldsymbol{\Theta} = \hat{\tilde{\mathbf{A}}}\mathbf{X}\boldsymbol{\Theta}.$$

The last step is based on the definition of the normalized Laplacian matrix $\tilde{\mathbf{L}} = \mathbf{I} - \hat{\tilde{\mathbf{A}}}$. We then obtain that when $K = 1$, we have

$$\mathbf{U}(\boldsymbol{\Theta}) = \hat{\tilde{\mathbf{A}}}\mathbf{X}\boldsymbol{\Theta} = \mathbf{U}^{(1)} = \mathbf{X}\hat{\boldsymbol{\Theta}}^{(1)} = \mathbf{X}\hat{\boldsymbol{\Theta}}^*.$$

Note that the orthogonal assumption of the node feature matrix $\mathbf{X}$ is generally easy to satisfy. If the graph does not have node features, we can use the one-hot node index vector to construct $\mathbf{X}$, which trivially makes $\mathbf{X}$ an orthogonal matrix. If the graph has node features, then we can normalize the node feature vector $x_i$ and apply the Gram-Schmidt process to generate an orthogonal $\mathbf{X}$ as the input.

*The Hypothesis Step.* When $K \geq 2$, meaning that we apply the gradient descent for $K$ times on Problem (24), we assume that,

$$\hat{\boldsymbol{\Theta}}^* = \hat{\boldsymbol{\Theta}}^{(K)} = \left(\mathbf{I} - \sum_{k=0}^{K-1} \mathbf{X}^{\mathsf{T}}\tilde{\mathbf{L}}\hat{\tilde{\mathbf{A}}}^k\mathbf{X}\right)\boldsymbol{\Theta}.$$

$$\mathbf{U}^{(K)} = \hat{\tilde{\mathbf{A}}}^K\mathbf{X}\boldsymbol{\Theta}.$$

Then, we immediately obtain,

$$\mathbf{U}(\boldsymbol{\Theta}) = \hat{\tilde{\mathbf{A}}}^K\mathbf{X}\boldsymbol{\Theta} = \mathbf{U}^{(K)} = \mathbf{X}\hat{\boldsymbol{\Theta}}^{(K)} = \mathbf{X}\hat{\boldsymbol{\Theta}}^*.$$

*The Inductive Step.* Consider the $K + 1$ case when we apply the gradient descent algorithm again based on $\hat{\boldsymbol{\Theta}}^{(K)}$. Similar to the base case, we now have,

$$\begin{aligned}
\hat{\boldsymbol{\Theta}}^* &= \hat{\boldsymbol{\Theta}}^{(K+1)} \\
&= \hat{\boldsymbol{\Theta}}^{(K)} - \alpha \cdot 2\mathbf{X}^{\mathsf{T}}\tilde{\mathbf{L}}\mathbf{U}^{(K)} \\
&= \left(\mathbf{I} - \sum_{k=0}^{K-1} \mathbf{X}^{\mathsf{T}}\tilde{\mathbf{L}}\hat{\tilde{\mathbf{A}}}^k\mathbf{X}\right)\boldsymbol{\Theta} - \mathbf{X}^{\mathsf{T}}\tilde{\mathbf{L}}\hat{\tilde{\mathbf{A}}}^K\mathbf{X}\boldsymbol{\Theta} \\
&= \left(\mathbf{I} - \sum_{k=0}^{K} \mathbf{X}^{\mathsf{T}}\tilde{\mathbf{L}}\hat{\tilde{\mathbf{A}}}^k\mathbf{X}\right)\boldsymbol{\Theta}.
\end{aligned}$$

The second to last step is based on the hypothesis. Also, for $\mathbf{U}^{(K+1)}$, we have,

$$\begin{aligned}
\mathbf{U}^{(K+1)} &= \mathbf{X}\hat{\boldsymbol{\Theta}}^{(K+1)} \\
&= \left(\mathbf{X} - \sum_{k=0}^{K} \mathbf{X}\mathbf{X}^{\mathsf{T}}\tilde{\mathbf{L}}\hat{\tilde{\mathbf{A}}}^k\mathbf{X}\right)\boldsymbol{\Theta} \\
&= \left(\mathbf{I} - \sum_{k=0}^{K} \tilde{\mathbf{L}}\hat{\tilde{\mathbf{A}}}^k\right)\mathbf{X}\boldsymbol{\Theta} \\
&= \left(\mathbf{I} - \tilde{\mathbf{L}} - \tilde{\mathbf{L}}\hat{\tilde{\mathbf{A}}} - \tilde{\mathbf{L}}\hat{\tilde{\mathbf{A}}}^2 - \cdots - \tilde{\mathbf{L}}\hat{\tilde{\mathbf{A}}}^K\right)\mathbf{X}\boldsymbol{\Theta} \\
&= \left(\hat{\tilde{\mathbf{A}}} - \tilde{\mathbf{L}}\hat{\tilde{\mathbf{A}}} - \tilde{\mathbf{L}}\hat{\tilde{\mathbf{A}}}^2 - \cdots - \tilde{\mathbf{L}}\hat{\tilde{\mathbf{A}}}^K\right)\mathbf{X}\boldsymbol{\Theta} \\
&= \left(\hat{\tilde{\mathbf{A}}}^2 - \tilde{\mathbf{L}}\hat{\tilde{\mathbf{A}}}^2 - \cdots - \tilde{\mathbf{L}}\hat{\tilde{\mathbf{A}}}^K\right)\mathbf{X}\boldsymbol{\Theta} \\
&= \cdots \\
&= \hat{\tilde{\mathbf{A}}}^{K+1}\mathbf{X}\boldsymbol{\Theta}.
\end{aligned}$$

Then, we immediately obtain,

$$\mathbf{U}(\mathbf{\Theta}) = \hat{\mathbf{A}}^{K+1}\mathbf{X}\mathbf{\Theta} = \mathbf{U}^{(K+1)} = \mathbf{X}\hat{\mathbf{\Theta}}^{(K+1)} = \mathbf{X}\hat{\mathbf{\Theta}}^*.$$

This concludes the proof. □

### E.2 Theorem 3.2

*Proof.* In the binary case, Eq. (5) is equivalent to finding the maximum a posteriori (MAP) estimate of a posterior probability distribution whose density function $\mathbb{P}(\mathbf{u}(\mathbf{\Theta}) \mid \mathbf{y})$ relates to the objective function via

$$\mathbb{P}(\mathbf{u}(\mathbf{\Theta}) \mid \mathbf{y}) \propto \exp\left(-\mathcal{L}\right).$$

Note that, we have,

$$\exp\left(-\mathcal{L}\right) = \exp\left(-\operatorname{Tr}\left(\mathbf{u}(\mathbf{\Theta})^\mathsf{T}\tilde{\mathbf{L}}_\lambda\mathbf{u}(\mathbf{\Theta})\right)\right)\exp\left(-\sum_{i\in\mathcal{V}_l}\ell(\sigma(u_i(\mathbf{\Theta})), y_i)\right)$$

$$= \exp\left(-\mathbf{u}(\mathbf{\Theta})^\mathsf{T}\tilde{\mathbf{L}}_\lambda\mathbf{u}(\mathbf{\Theta})\right)\exp\left(-\sum_{i\in\mathcal{V}_l}\ell(\sigma(u_i(\mathbf{\Theta})), y_i)\right)$$

$$\propto (2\pi)^{-n/2}\det(\tilde{\mathbf{L}}_\lambda^{-1})^{-1/2}\exp\left(-\frac{1}{2}(\mathbf{u}(\mathbf{\Theta}) - \mathbf{0})^\mathsf{T}(\tilde{\mathbf{L}}_\lambda^{-1})^{-1}(\mathbf{u}(\mathbf{\Theta}) - \mathbf{0})\right)\exp\left(-\sum_{i\in\mathcal{V}_l}\ell(\sigma(u_i(\mathbf{\Theta})), y_i)\right)$$

$$= \mu(\mathbf{u}(\mathbf{\Theta}))\exp\left(-\Phi_\ell(\mathbf{u}(\mathbf{\Theta}))\right).$$

Here, the prior $\mu(\mathbf{u}(\mathbf{\Theta}))$ follows a Gaussian prior $\mathcal{N}(\mathbf{0}, \tilde{\mathbf{L}}_\lambda^{-1})$ and the likelihood $\exp\left(-\Phi_\ell(\mathbf{u}(\mathbf{\Theta}))\right)$ is defined by the likelihood potential $\Phi_\ell(\mathbf{u}(\mathbf{\Theta})) := \sum_{i\in\mathcal{V}_l}\ell(\sigma(u_i(\mathbf{\Theta})), y_i)$. $\mathbf{y} = [y_i]_{i\in\mathcal{V}_l} \in \{0,1\}^{n_l}$. □

### E.3 Theorem 3.3

We first revisit Laplace Approximation. Laplace approximation is a popular technique for approximating non-Gaussian distributions with a Gaussian distribution. A given probability distribution, identified by its probability density function (PDF) $\mathbb{P}(\mathbf{x})$ can be approximated via another Gaussian distribution as follows.

$$\mathbf{x} \sim \mathcal{N}(\hat{\mathbf{x}}, \hat{\mathbf{C}}), \quad \hat{\mathbf{x}} = \arg\max_{\mathbf{x}\in\mathbb{R}^n}\mathbb{P}(\mathbf{x}), \quad \hat{\mathbf{C}} = \left(-\nabla^2\ln\left(\mathbb{P}(\mathbf{x})\right)|_{\mathbf{x}=\hat{\mathbf{x}}}\right)^{-1}.$$

Here, $\hat{\mathbf{x}}$ is the MAP estimator of $\mathbb{P}(\mathbf{x})$ and $\hat{\mathbf{C}}$ is the Hessian matrix of the negative-log density of the distribution evaluated at the MAP estimator $\hat{\mathbf{x}}$.

*Proof.* According to the Laplace Approximation, we know

$$\boldsymbol{\alpha} \mid \mathbf{y} \sim \mathcal{N}(\hat{\boldsymbol{\alpha}}, \hat{\mathbf{C}}_{\hat{\boldsymbol{\alpha}}}), \quad \hat{\boldsymbol{\alpha}} = \arg\min_{\boldsymbol{\alpha}}\tilde{\mathcal{L}}.$$

Then it is easy to verify that

$$\nabla_{\boldsymbol{\alpha}}\tilde{\mathcal{L}} = \mathbf{\Lambda}_\lambda\boldsymbol{\alpha} + \sum_{i\in\mathcal{V}_l}F(\sigma(\mathbf{e}_i^\mathsf{T}\mathbf{V}\hat{\boldsymbol{\alpha}}), y_i)\mathbf{V}^\mathsf{T}\mathbf{e}_i = \mathbf{\Lambda}_\lambda\boldsymbol{\alpha} + \mathbf{V}^\mathsf{T}\sum_{i\in\mathcal{V}_l}F(\sigma(\mathbf{e}_i^\mathsf{T}\mathbf{V}\hat{\boldsymbol{\alpha}}), y_i)\mathbf{e}_i.$$

$$\nabla_{\boldsymbol{\alpha}}^2\tilde{\mathcal{L}} = \mathbf{\Lambda}_\lambda + \mathbf{V}^\mathsf{T}(\sum_{i\in\mathcal{V}_l}F'(\sigma(\mathbf{e}_i^\mathsf{T}\mathbf{V}\hat{\boldsymbol{\alpha}}), y_i)\mathbf{e}_i\mathbf{e}_i^\mathsf{T})\mathbf{V} = \mathbf{\Lambda}_\lambda + \mathbf{V}^\mathsf{T}\left(\sum_{i\in\mathcal{V}_l}F'(\sigma(\mathbf{e}_i^\mathsf{T}\mathbf{V}\hat{\boldsymbol{\alpha}}), y_i)\mathbf{e}_i\mathbf{e}_i^\mathsf{T}\right)\mathbf{V}.$$

Therefore, we have,

$$\hat{\mathbf{C}}_{\hat{\boldsymbol{\alpha}}} = (\mathbf{\Lambda}_\lambda + \mathbf{V}^\mathsf{T}\left(\sum_{i\in\mathcal{V}_l}F'(\sigma(\mathbf{e}_i^\mathsf{T}\mathbf{V}\hat{\boldsymbol{\alpha}}), y_i)\mathbf{e}_i\mathbf{e}_i^\mathsf{T}\right)\mathbf{V})^{-1}.$$

We define $F(x,y) := \frac{\partial\ell}{\partial x}(x,y)$, $F'(x,y) := \frac{\partial^2\ell}{\partial x^2}(x,y)$. $\mathbf{e}_i$ represents the $i$-th standard basis vector. □

By applying the Laplace approximation to the non-Gaussian posterior distributions of $\mathbb{P}(\mathbf{u}(\boldsymbol{\Theta}) \mid \mathbf{y})$, we can approximate look-ahead updates for calculating the designed acquisition function.

## E.4 Theorem 3.4

*Proof.* Starting with the current MAP estimator $\hat{\boldsymbol{\alpha}}$, we have,

$$
\begin{aligned}
\hat{\boldsymbol{\alpha}}^{k,y_k^+} &= \hat{\boldsymbol{\alpha}} - \left(\nabla^2 \mathcal{L}^{k,y_k^+}\right)^{-1}\left(\nabla \mathcal{L}^{k,y_k^+}\right) \\
&= \hat{\boldsymbol{\alpha}} - \left(\hat{\mathbf{C}}_{\hat{\boldsymbol{\alpha}}}^{-1} + F'(\sigma(\mathbf{e}_k^\mathsf{T}\mathbf{V}\hat{\boldsymbol{\alpha}}), y_k^+)\mathbf{v}^k\mathbf{v}^{k\mathsf{T}}\right)^{-1}\left(\nabla \mathcal{L} + F(\sigma(\mathbf{e}_k^\mathsf{T}\mathbf{V}\hat{\boldsymbol{\alpha}}), y_k^+)(\mathbf{e}_k^\mathsf{T}\mathbf{V})^\mathsf{T}\right) \\
&= \hat{\boldsymbol{\alpha}} - \left(\mathbf{C}_{\hat{\boldsymbol{\alpha}}} - \mathbf{C}_{\hat{\boldsymbol{\alpha}}}(\mathbf{e}_k^\mathsf{T}\mathbf{V})^\mathsf{T}\left(\frac{1}{F'(\sigma(\mathbf{e}_k^\mathsf{T}\mathbf{V}\hat{\boldsymbol{\alpha}}), y_k^+)} + (\mathbf{e}_k^\mathsf{T}\mathbf{V})\mathbf{C}_{\hat{\boldsymbol{\alpha}}}(\mathbf{e}_k^\mathsf{T}\mathbf{V})^\mathsf{T}\right)^{-1}(\mathbf{v}^k)^\mathsf{T}\mathbf{C}_{\hat{\boldsymbol{\alpha}}}\right)F(\sigma(\mathbf{e}_k^\mathsf{T}\mathbf{V}\hat{\boldsymbol{\alpha}}), y_k^+)(\mathbf{e}_k^\mathsf{T}\mathbf{V})^\mathsf{T} \\
&= \hat{\boldsymbol{\alpha}} - \frac{F(\sigma(\mathbf{e}_k^\mathsf{T}\mathbf{V}\hat{\boldsymbol{\alpha}}), y_k^+)}{1 + F'(\sigma(\mathbf{e}_k^\mathsf{T}\mathbf{V}\hat{\boldsymbol{\alpha}}), y_k^+)(\mathbf{e}_k^\mathsf{T}\mathbf{V})\hat{\mathbf{C}}_{\hat{\boldsymbol{\alpha}}}(\mathbf{e}_k^\mathsf{T}\mathbf{V})^\mathsf{T}}\hat{\mathbf{C}}_{\hat{\boldsymbol{\alpha}}}(\mathbf{e}_k^\mathsf{T}\mathbf{V})^\mathsf{T}
\end{aligned}
$$

Note that we define $F(x,y) := \frac{\partial \ell}{\partial x}(x,y)$, $F'(x,y) := \frac{\partial^2 \ell}{\partial x^2}(x,y)$. $\mathbf{e}_i$ represents the $i$-th standard basis vector. $\square$

## E.5 Theorem 3.5

*Proof.* We focus on the regression task in this theorem. The non-linear activation functions can now be removed and the node embedding $\mathbf{u}(\boldsymbol{\Theta})$ can directly be used as the output prediction $\hat{\mathbf{y}} = \mathbf{u}(\boldsymbol{\Theta}) \in \mathbb{R}^n$ and compared with the ground-truth label $\mathbf{y} \in \mathbb{R}^n$.

Since the squared loss $\ell(x,y) = \frac{1}{2}(x-y)^2$ is used, we can easily verify that,

$$F(x,y) = \frac{\partial \ell}{\partial x}(x,y) = x - y \tag{25}$$

$$F'(x,y) = \frac{\partial^2 \ell}{\partial^2 x}(x,y) = 1. \tag{26}$$

Since the task is changed into the regression task, the acquisition function Eq. (11) is now changed as follows.

$$
\begin{aligned}
\mathcal{A}(k) &= \int_{-\infty}^{+\infty} \mathbb{P}(y_k^+|k)\left|\frac{\mathbf{v}_k^\mathsf{T}\hat{\boldsymbol{\alpha}} - y_k^+}{1 + \mathbf{v}_k^\mathsf{T}\hat{\mathbf{C}}_{\hat{\boldsymbol{\alpha}}}\mathbf{v}_k}\right|\|\hat{\mathbf{C}}_{\hat{\boldsymbol{\alpha}}}\mathbf{v}_k\|_2 dy_k^+ \\
&= \frac{1}{1 + \mathbf{v}_k^\mathsf{T}\hat{\mathbf{C}}_{\hat{\boldsymbol{\alpha}}}\mathbf{v}_k}\|\hat{\mathbf{C}}_{\hat{\boldsymbol{\alpha}}}\mathbf{v}_k\|_2 \int_{-\infty}^{+\infty} \mathbb{P}(y_k^+|k)|\mathbf{v}_k^\mathsf{T}\hat{\boldsymbol{\alpha}} - y_k^+|dy_k^+.
\end{aligned}
$$

Note that we use the prediction by the current model to approximate $\mathbb{P}(y_k^+|k) = \mathbf{e}_k^\mathsf{T}\mathbf{u}(\boldsymbol{\Theta}) = \mathbf{v}_k^\mathsf{T}\hat{\boldsymbol{\alpha}}$. Hence, we have,

$$
\begin{aligned}
\mathcal{A}(k) &= \frac{1}{1 + \mathbf{v}_k^\mathsf{T}\hat{\mathbf{C}}_{\hat{\boldsymbol{\alpha}}}\mathbf{v}_k}\|\hat{\mathbf{C}}_{\hat{\boldsymbol{\alpha}}}\mathbf{v}_k\|_2 \int_{-\infty}^{+\infty} \mathbf{v}_k^\mathsf{T}\hat{\boldsymbol{\alpha}}|\mathbf{v}_k^\mathsf{T}\hat{\boldsymbol{\alpha}} - y_k^+|dy_k^+ \\
&= \frac{1}{1 + \mathbf{v}_k^\mathsf{T}\hat{\mathbf{C}}_{\hat{\boldsymbol{\alpha}}}\mathbf{v}_k}\|\hat{\mathbf{C}}_{\hat{\boldsymbol{\alpha}}}\mathbf{v}_k\|_2 \left\{\int_{-\infty}^{\mathbf{v}_k^\mathsf{T}\hat{\boldsymbol{\alpha}}} \mathbf{v}_k^\mathsf{T}\hat{\boldsymbol{\alpha}}(\mathbf{v}_k^\mathsf{T}\hat{\boldsymbol{\alpha}} - y_k^+)dy_k^+ + \int_{\mathbf{v}_k^\mathsf{T}\hat{\boldsymbol{\alpha}}}^{+\infty} \mathbf{v}_k^\mathsf{T}\hat{\boldsymbol{\alpha}}(y_k^+ - \mathbf{v}_k^\mathsf{T}\hat{\boldsymbol{\alpha}})dy_k^+\right\} \\
&= \frac{1}{1 + \mathbf{v}_k^\mathsf{T}\hat{\mathbf{C}}_{\hat{\boldsymbol{\alpha}}}\mathbf{v}_k}\|\hat{\mathbf{C}}_{\hat{\boldsymbol{\alpha}}}\mathbf{v}_k\|_2\|\hat{\mathbf{C}}_{\hat{\boldsymbol{\alpha}}}\mathbf{v}_k\|_2 \\
&= \frac{1}{1 + \mathbf{v}_k^\mathsf{T}\hat{\mathbf{C}}_{\hat{\boldsymbol{\alpha}}}\mathbf{v}_k}\|\hat{\mathbf{C}}_{\hat{\boldsymbol{\alpha}}}\mathbf{v}_k\|_2^2.
\end{aligned}
$$

This completes the proof of Eq. (12).

Table 7: Summary of five datasets used in the experiments.

| Dataset | #Nodes | #Features | #Edges | #Classes | #Train/Val/Test | Setting | Type |
|---|---|---|---|---|---|---|---|
| Cora | 2,708 | 1,433 | 5,429 | 7 | 1,208/500/1,000 | Transductive | Citation Network |
| Citeseer | 3,327 | 3,703 | 4,732 | 6 | 1,827/500/1,000 | Transductive | Citation Network |
| PubMed | 19,717 | 600 | 44,338 | 3 | 18,217/500/1,000 | Transductive | Citation Network |
| Reddit | 232,965 | 602 | 11,606,919 | 41 | 155,310/23,297/54,358 | Inductive | Social Network |
| ogbn-arxiv | 169,343 | 128 | 1,166,243 | 40 | 90,941/29,799/48,603 | Transductive | Citation Network |

Next, we have,

$$
\mathbb{E}_{i \in \mathcal{V}_u^{(t)} \setminus \{k\}}[\ell(\hat{y}_i, y_i)]
$$

$$
= \mathbb{E}\left( \sum_{i \in \mathcal{V}_u^{(t)} \setminus \{k\}} \ell(\hat{y}_i, y_i) \right)
$$

$$
= \mathbb{E}\left( (\mathbf{u}^{k,y_k^+}(\boldsymbol{\Theta}) - \mathbf{y})^\intercal (\mathbf{u}^{k,y_k^+}(\boldsymbol{\Theta}) - \mathbf{y}) \right)
$$

$$
= \mathbb{E}\left( \mathrm{Tr}\left( (\mathbf{u}^{k,y_k^+}(\boldsymbol{\Theta}) - \mathbf{y})(\mathbf{u}^{k,y_k^+}(\boldsymbol{\Theta}) - \mathbf{y})^\intercal \right) \right)
$$

$$
= \mathrm{Tr}\left( \mathbb{E}\left( (\mathbf{u}^{k,y_k^+}(\boldsymbol{\Theta}) - \mathbf{y})(\mathbf{u}^{k,y_k^+}(\boldsymbol{\Theta}) - \mathbf{y})^\intercal \right) \right)
$$

$$
= \mathrm{Tr}(\mathrm{var}(\mathbf{u}^{k,y_k^+}(\boldsymbol{\Theta}))).
$$

When the loss is set as squared loss, we can modify the functions regarding the truncated spectral projection. Recall that $\boldsymbol{\alpha}|\mathbf{y} \sim \mathcal{N}(\hat{\boldsymbol{\alpha}}, \hat{\mathbf{C}}_{\hat{\boldsymbol{\alpha}}})$ with $\hat{\mathbf{C}}_{\hat{\boldsymbol{\alpha}}} = (\boldsymbol{\Lambda}_\lambda + \mathbf{V}^\intercal \mathbf{P}^\intercal \mathbf{P} \mathbf{V})^{-1}$ and $\hat{\boldsymbol{\alpha}} = \hat{\mathbf{C}}_{\hat{\boldsymbol{\alpha}}} \mathbf{V}^\intercal \mathbf{P}^\intercal \mathbf{y}$ so that we have $\mathbf{u}(\boldsymbol{\Theta}) \sim \mathcal{N}(\mathbf{V}\hat{\boldsymbol{\alpha}}, \mathbf{V}\hat{\mathbf{C}}_{\hat{\boldsymbol{\alpha}}} \mathbf{V}^\intercal)$. Therefore, we know,

$$
\mathbb{E}_{i \in \mathcal{V}_u^{(t)} \setminus \{k\}}[\ell(\hat{y}_i, y_i)]
$$

$$
= \mathrm{Tr}\left( \mathrm{var}(\mathbf{u}^{k,y_k^+}(\boldsymbol{\Theta})) \right)
$$

$$
= \mathrm{Tr}\left( \mathbf{V}\hat{\mathbf{C}}_{\hat{\boldsymbol{\alpha}}}^{k,y_k^+} \mathbf{V}^\intercal \right)
$$

$$
= \mathrm{Tr}\left( \mathbf{V}\left( \hat{\mathbf{C}}_{\hat{\boldsymbol{\alpha}}} - \frac{1}{1 + \mathbf{v}_k^\intercal \hat{\mathbf{C}}_{\hat{\boldsymbol{\alpha}}} \mathbf{v}_k} \hat{\mathbf{C}}_{\hat{\boldsymbol{\alpha}}} \mathbf{v}_k \mathbf{v}_k^\intercal \hat{\mathbf{C}}_{\hat{\boldsymbol{\alpha}}} \right) \mathbf{V}^\intercal \right)
$$

$$
= C - \frac{1}{1 + \mathbf{v}_k^\intercal \hat{\mathbf{C}}_{\hat{\boldsymbol{\alpha}}} \mathbf{v}_k} \| \hat{\mathbf{C}}_{\hat{\boldsymbol{\alpha}}} \mathbf{v}_k \|_2^2.
$$

Here, $C$ is some constant irrelevant to $k$. Hence, we immediately know,

$$
\arg\min_{k \in \mathcal{V}_u^{(t)}} \mathbb{E}_{i \in \mathcal{V}_u^{(t)} \setminus \{k\}}[\ell(\hat{y}_i, y_i)] = \arg\max_{k \in \mathcal{V}_u^{(t)}} \frac{1}{1 + \mathbf{v}_k^\intercal \hat{\mathbf{C}}_{\hat{\boldsymbol{\alpha}}}} \| \hat{\mathbf{C}}_{\hat{\boldsymbol{\alpha}}} \mathbf{v}_k \|_2^2. \tag{27}
$$

$\square$

## F  Experimental Setup

Most of the experimental settings follow the existing work [90] for graph active learning.

### F.1  Datasets Description

We summarize the statistics of five datasets in Table 7. The detailed introductions of each dataset are presented as follows.

**Cora, Citeseer, PubMed**  Cora, Citeseer, and PubMed [1] are three popular citation network datasets and we follow the train/validation/test split in the original GCN paper [32]. In these three datasets,

---

[1] `https://github.com/tkipf/gcn/tree/master/gcn/data`

papers from different topics are considered nodes, and the edges are citation relationships among the papers. The node attributes are binary word vectors, and class labels are the topics that papers belong to.

**Reddit** Reddit is a social network dataset obtained from the community structure of online Reddit posts. Reddit is a large online discussion forum where users post and comment on content in different topical communities. We predict which community different Reddit posts belong to. In total, this dataset contains 232,965 posts with an average degree of 492.

**ogbn-arxiv** ogbn-arxiv [2] is a directed graph, representing the citation network between all Computer Science (CS) arXiv papers. Each node is an arXiv paper and each directed edge indicates that one paper cites another one. Each paper comes with a 128-dimensional feature vector obtained by averaging the embeddings of words in its title and abstract. The embeddings of individual words are computed by running the skip-gram model. The task is to predict the 40 subject areas of arXiv CS papers, e.g., cs.AI, cs.LG, and cs.OS, which are manually determined (i.e., labeled) by the paper's authors and arXiv moderators.

### F.2 Implementation Details

All the experiments are conducted in the batch active learning setting unless otherwise specified. For simplicity, we select the size of the query node set as $b = 5$ in one active learning iteration for all baseline models by default. In Algorithm 3, we reduce the pool of the unlabeled nodes from $\mathcal{V}_u^{(t)}$ to $\mathcal{S} \subset \mathcal{V}_u^{(t)}$. To be more specific, we sample $10\%$ of the nodes in the current pool of unlabeled nodes $\mathcal{V}_u^{(t)}$ uniformly at random so that $|\mathcal{S}| = 0.1|\mathcal{V}_u^{(t)}|$. The query set comprises the top $b = 5$ maximizers of the designed acquisition function on all nodes in $\mathcal{S}$. Note that our experiments have a simple assumption about an error-less oracle (e.g., by human experts) since our main contributions focus on effectiveness measurement and node selection. In all the experiments in this work, we simulate this perfect oracle by recording all the ground-truth labels for all the unlabeled nodes into the database. This database serves as the error-less oracle and handles the query requests made by the active learning model. Therefore, no real human annotators are involved in our experiments without leading to any ethical concerns. However, in real-world applications, either domain experts or crowd-sourcing services are allowed in the active learning process, making the existence of an error-less oracle almost impossible. Handling noisy oracles is orthogonal to our work, we leave it for future work.

The hardware configurations are four GeForce RTX 2080 Ti GPUs and SuperServer Dual Intel Xeon CPU (Ten-core 2.20GHz) Processor. All the implementation details of the baselines and the proposed method are discussed below.

For **AGE** [4], to procure well-trained models and guarantee that their model-based selection criteria work well, GCN is trained for 50 epochs in each node selection iteration. AGE is implemented with its open-source version.

For **ALG** [88], we follow the public code with the original paper. More specifically, we choose the ALG with approximate QBC, where we use an MLP operating on the $K$-hop averaged features to approximate a $K$-layer GCN. We also adjust the training budget for the AL iterations accordingly.

For **RIM** [89], we fix the threshold as 0.01 and keep all the other settings unchanged in the original RIM implementation.

For **IGP** [90], we fix the key hyperparameter $\alpha$ as 2 and set the degree for dismissing uninfluential nodes as 11. Other hyper-parameters are all identical to the open-source code.

For **GraphPart** [41], we stick to the original version of GraphPart with the K-Medoids algorithm as the clustering method. All the hyperparameters are set as default in the provided implementation.

For **GEEM** [51] in the efficiency comparison, we choose the proposed GEEM model based on SGC and expected error minimization without the preemptive for a fair comparison.

For our proposed **DOCTOR** method, we set the backbone GNN model as the SGC [68]. We tune all the hyperparameters via grid search. The best model for each combination is saved based on

---

[2]`https://ogb.stanford.edu/docs/nodeprop/\#ogbn-arxiv`

Table 8: Search space for the main hyperparameters used in the proposed DOCTOR method.

| Hyperparameters | Search Range |
|---|---|
| Number of layers | {1,2,3,4} |
| Hidden dimensions | {64,128,256,512} |
| Dropout rate | {0.4,0.5,0.6} |
| Training epochs | {50,100,200} |
| Weight decay | {1e-7,1e-6,1e-5,5e-4,1e-4,5e-3} |
| Learning rate | {0.001,0.01,0.1} |
| $m$ | {10,20,50,100,200} |
| $\lambda$ | {1e-4,5e-4,1e-3,5e-3,1e-2,5e-2,1e-1} |

validation performance and is applied to the test set. For the hyperparameters in the backbone SGC model, the default hyperparameter settings are as follows. The number of layers is set as 2. We use 128 hidden dimensions in the intermediate layer. The SGC layers are equipped with skip connection, sum aggregation, batch normalization, and a dropout rate of 0.5. The default learning rate is 0.01 and we train the model for 200 epochs. The optimizer is set as Adam with a weight decay of 5e-4. For the acquisition function proposed in the **DOCTOR** algorithm, we have two key hyperparameters. One is the number of the smallest eigenvalues $m$ of the graphs' normalized Laplacian matrix and the other is the balancing factor $\lambda$ in $\boldsymbol{\Lambda}_\lambda$. Due to the sparse nature of the graphs, we choose to consider $m = 50$ smallest eigenvalues of $\tilde{\mathbf{L}}_\lambda$ in the default setting. We exploit the Lanczos algorithm [3] with the help of the existing implementation [4] to obtain the lowest eigenvalues and corresponding eigenvectors in good precision. For the balancing factor in $\tilde{\mathbf{L}}_\lambda$, we set $\lambda = 5e - 3$ by default. The search ranges of the hyperparameters are reported in Table 8.

# G    Experimental Results

## G.1    Efficiency

We first present a formal analysis of the time complexity of our proposed DOCTOR method. For simplicity, we analyze Algorithm 3 with the batch AL setting for the binary node classification task. Then we compare DOCTOR with another performance-based method called GEEM [51], which extends the expected error minimization principle for graph AL. We investigate the trade-off between prediction accuracy and running time through empirical experiments.

### G.1.1    Time Complexity Analysis

We focus on Algorithm 3 for time complexity analysis. More specifically, we only need to consider the time complexity of one AL iteration (lines 5-12), which is the main overhead of the AL methods. Note that, like many other AL methods, DOCTOR also uses the backbone model for training, whose complexity is irrelevant to the design of the acquisition function. We let the complexity of training the backbone model or the SGC model in Algorithm 3, as $O(\mathcal{M})$ (line 5). For a fair comparison, we will also use the same backbone model for training in other baselines when conducting empirical experiments. This portion of the computational overhead due to the training of the backbone model is fixed in each AL loop. Note that DOCTOR has some preprocessing steps in line 2 before the actual AL iterations. Thanks to some existing techniques, the computational overhead regarding the truncated spectral projection will not affect the total cost too much. Since the Laplacian matrix is symmetric $\tilde{\mathbf{L}}_\lambda = \tilde{\mathbf{L}}_\lambda^\intercal$, we can apply the famous Lanczos algorithm on this Hermitian matrix to find the $m$-lowest eigenvalues and corresponding eigenvectors so that $\mathbf{V}$ and $\boldsymbol{\Lambda}_\lambda$ can be composited accordingly. It is well known that the Lanczos algorithm requires roughly $O(mn)$ time to evaluate the extreme eigenvalues and eigenvectors (line 2). We will see that this extra computational cost is negligible when compared to the overhead in one AL iteration.

We focus on one AL iteration (lines 5-12) in Algorithm 3. Training an SGC model takes $O(\mathcal{M})$ time (line 5). To compute $\hat{\boldsymbol{\alpha}}$, it costs $O(mn)$ through the matrix-vector multiplication. We can get $\hat{\mathbf{C}}_{\hat{\boldsymbol{\alpha}}}$ via Eq. (9) in $O(m^3)$ due to the matrix inversion of size $m \times m$. So we have $O(mn + m^3)$ in total

---

[3]https://en.wikipedia.org/wiki/Lanczos_algorithm
[4]https://spectralib.org

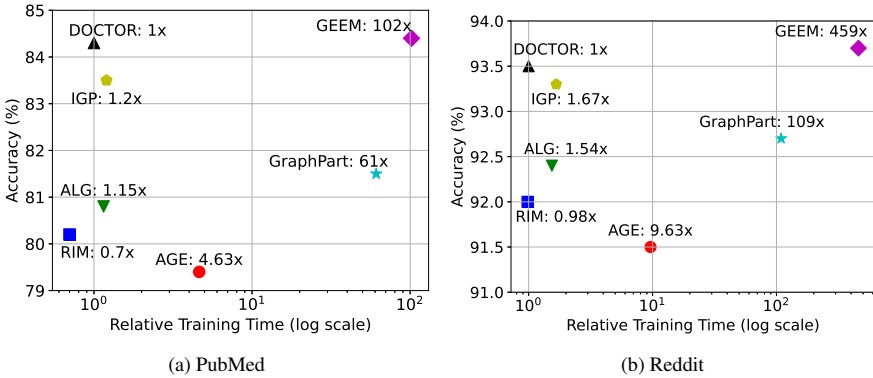

(a) PubMed                    (b) Reddit

Figure 7: Prediction accuracy versus training time of baselines and DOCTOR on two datasets.

for line 6. Assuming the size of $\mathcal{S}$ is fixed as $|\mathcal{S}|$, it is easy to check the overhead for computing the expected model change (lines 8-10) via Eq. (11) is $O(|\mathcal{S}|m^3)$ since several intermediate variables like $\hat{\mathbf{C}}_{\hat{\alpha}}$ have been computed. To select the final top-$b$ nodes as the query set, we apply the QuickSelect algorithm to get the threshold $b$-largest acquisition function value in $O(|\mathcal{S}|)$ and then filter out all nodes that have acquisition function value greater than the threshold value. Hence, it takes $O(|\mathcal{S}|)$ time to select top $b$ maximizes (line 11). To sum up, it takes $O(\mathcal{M} + mn + m^3 + |\mathcal{S}|m^3 + |\mathcal{S}|)$ to complete one AL iteration. Since we have $1 < m \ll n$ and $|\mathcal{S}| > 1$ in the batch AL setting, we know that it costs $O(\mathcal{M} + mn + |\mathcal{S}|m^3)$ in one AL iteration (lines 5-12) in Algorithm 3. Note that this result is a great improvement compared with the time complexity of GEEM [51] ($O(|\mathcal{S}|\mathcal{M})$), where we have to re-train the backbone model for at least $|\mathcal{S}|$ times to evaluate the performance when each candidate node is added into the labeled node set in one single AL iteration.

### G.1.2 Running Time Comparison

In Figure 7, we plot the performance of the state-of-the-art baseline models over their end-to-end training time relative to that of DOCTOR on the PubMed and Reddit dataset. From Figure 7, we can observe that DOCTOR is much more efficient than the other performance-based method, GEEM, without any significant loss in terms of the prediction accuracy. Meanwhile, it is almost as efficient as RIM, which is the most efficient method of all, but DOCTOR achieves a much higher prediction accuracy. Therefore, our proposed method strikes a good balance between performance and efficiency.

### G.2 Generalization to Other GNNs

We further verify the generalization ability of the proposed acquisition function in DOCTOR when the backbone model is changed to other GNNs. The results on the inductive dataset, Reddit, are summarized in Table 9. We make several key observations from the results in Table 9. First, for the fixed backbone model, DOCTOR indeed achieves the best performance even though its improvement from the second-best is often marginal since DOCTOR is designed for a transductive setting. Therefore, the proposed acquisition function generalizes well to other GNNs because our Bayesian learning framework can incorporate many GNNs based on the unified optimization framework, discussed in Appendix C.2.3. Second, for the fixed method, the use of any other GNNs, like SGC, GCN, and APPNP, does not make much difference but the use of GraphSAGE will result in a significant improvement because GraphSAGE is specifically designed for the inductive setting. We leave the investigation of a specific design of graph active learning methods under the inductive setting for further work since it is out of the scope of this paper. For now, we can use GraphSAGE or other inductive GNNs as the backbone models.

### G.3 Sensitivity

We find it difficult to perform an ablation study on our proposed DOCTOR method since all the components designed in DOCTOR are correlated to each other. Therefore, we analyze the influence of key hyperparameters in DOCTOR, the number of top-$m$ smallest eigenvalues in the truncated

Table 9: Test accuracy (%) of different methods with different backbone GNNs on the Reddit dataset.

| Method | SGC | GCN | APPNP | GraphSAGE |
|---|---|---|---|---|
| Random | 91.3±0.5 | 91.1±0.5 | 91.4±0.6 | 94.3±0.4 |
| AGE | 91.5±0.4 | 91.6±0.3 | 92.0±0.7 | 94.8±0.4 |
| ALG | 92.4±0.6 | 92.4±0.3 | 92.8±0.6 | 95.2±0.5 |
| RIM | 92.0±0.7 | 92.1±0.7 | 92.3±0.9 | 95.0±0.8 |
| IGP | 93.3±0.4 | 93.4±0.2 | 93.6±0.5 | 95.7±0.7 |
| GraphPart | 92.7±1.0 | 92.5±0.8 | 92.8±0.7 | 94.7±0.6 |
| DOCTOR | **93.5±0.5** | **93.5±0.7** | **93.7±0.6** | **95.8±0.8** |

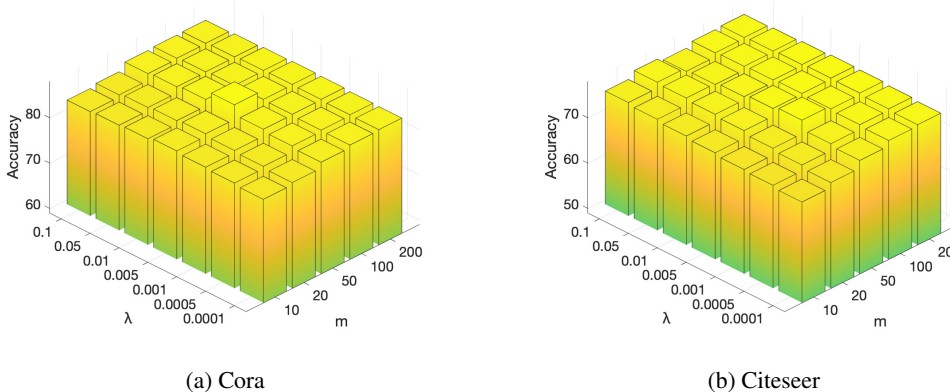

(a) Cora            (b) Citeseer

Figure 8: Influence of hyperparameters $\lambda$ and $m$ in the proposed DOCTOR method on two datasets.

spectral projection, and the balancing factor $\lambda$ in $\mathbf{\Lambda}_\lambda$. From Figure 8, we can see that the performance of DOCTOR is quite stable under different configurations of $\lambda$ and $m$. A larger $m$ usually leads to better performance since the approximation of the node embeddings projection will be more precise, but the improvement may be quite marginal if $m$ is too large. In addition, $\lambda$ should be carefully chosen to achieve optimal performance by avoiding setting it too small or too large.

## H  Broader Impacts

To avoid ethical concerns, our method can gain access to the ground-truth labeled database to simulate querying a human annotator in the experiments. DOCTOR can be employed in graph-related applications in the real world, like predicting malicious accounts with money laundry activities on transaction networks. However, when human annotators are involved, DOCTOR will face the noisy label issue and the information leakage concern. We encourage researchers to be aware of the limitations and privacy concerns of DOCTOR when deployed in real-world applications.