# OpenReview forum: "No Change, No Gain: Empowering Graph Neural Networks with Expected Model Change Maximization for Active Learning"
_NeurIPS.cc/2023/Conference — NeurIPS 2023 spotlight_

### Official Review · Reviewer_3rHV · 2023-06-30

**Soundness:** 3 good
**Presentation:** 3 good
**Contribution:** 3 good
**Rating:** 7
**Confidence:** 5

**Summary:**

This work proposes a novel active learning method for graph neural networks, extending the classic expected model change maximization method in the general active learning setting. The proposed method starts from a new Bayesian interpretation of the SGC model and unifies the training process of GNN models, including both forward and backward passes. Then the authors analyze the challenges of applying EMCM on graphs and use some approximation techniques for efficient computation. Finally, the connection between the proposed method and the expected error minimization method is revealed from the theoretical aspect.

**Strengths:**

1.	Originality: The paper is novel in general, with new EMCM method adaption on graphs and new GNN training understanding from the Bayesian view. The paper is the first work to extend the classic expected model change maximization method in general active learning literature to graph-structured data. The new unified Bayesian view of the GNN training process is also interesting and enlightening to the field. Previous works only consider the feedforward pass of the training when trying to unify the GNNsf from the optimization perspective.
2.	Quality: The quality of this work is above average, with a clear motivation and an interesting theoretical analysis of the method. First, the proposed method is strongly motivated by the Bayesian view of the GNNs from the theoretical perspective. Theorem 3.2 reveals the Bayesian interpretation of the general GNN training process, which lays the foundation of the following proposed method. Second, the approximated expected model change in terms of node embeddings can be solved by an elegant closed-form solution in Eq.(11) without re-training the model, which is quite efficient after employing some well-known simple techniques in the spectral graph theory. Third, the paper also theoretically proves that the proposed DOCTOR algorithm will be reduced to the EEM method when some simple assumptions are made, showing that DOCTOR equivalently reduces the expected test error as well, just like the EEM method. Forth, the experiments are convincing, with SOTA baselines (most of them are after 2022) regarding both accuracy and efficiency comparison.
3.	Clarity: The reviewer finds the paper easy to follow, perhaps since the reviewer is familiar with the related major references in both active learning and GNNs. The overall presentation of the paper is fairly ok, with clear notations and concise statements of the theorems. The case study in the experiments is also illustrative and supports Theorem 3.5 in a more concrete and empirical manner.
4.	Significance: The reviewer thinks this paper makes some contributions to both the active learning community and the GNN community since it connects these two domains and solves a significant problem, that is, how to adapt the classic active learning algorithm to GNNs. This work chooses the expected model change maximization algorithm and solves the potential challenges of extending EMCM on graphs effectively. For researchers in the AL domain, this work provides a new adaption of EMCM specifically designed for GNNs. For researchers in the GNN domain, this work also presents a performance-based AL method for GNNs since most of the current graph active learning literature only focus on uncertainty-based or information-density-based methods without good theoretical interpretation.


**Weaknesses:**

There are several potential improvements in this paper.

1.	The background knowledge of EMCM method in general active learning can be elaborated. The details behind Eq.(7) should be provided to give readers without active learning backgrounds more context. For example, the intuition behind the design of Eq.(7) can be discussed.

2.	For the efficiency comparison, only three small-scale datasets are used. It is suggested to test the running time performance of the proposed method and other baselines on the large-scale dataset like the ogbn dataset.

3.	The conclusion part is short. More future work can be added.

4.	Some typos should be fixed. For example, in line 219, "embedding s" should be "embeddings".

5.	Other comments in questions and limitations.


**Questions:**

1.	The current setting is only designed for node classification tasks. What about link-level and graph-level tasks? The reviewer knows that this question may be out of the scope of this work, but the reviewer is curious about the potential of extending this work to tasks like link predictions. Can we directly use the DOCTOR algorithm? If not, what are the challenges?
2.	The designed algorithm uses a simple sampling algorithm when dealing with the batch active learning setting. What are the possibilities of using other more advanced techniques to choose the query nodes in batches?


**Limitations:**

The reviewer generally agrees with the limitations discussed in Appendix H. This work currently only supports the batch active learning setting with a simple extended version of the sequential active learning solution. However, it is well-known that such a simple extension ignores the correlations between each query node in one batch, as they may reside within the same cluster in the embedding space.

---

> ### Author Rebuttal · Authors · 2023-08-08
>
> Thanks for your insightful review! We are glad to know that you found our paper novel, sound, clear, and significant. We hope these responses will address your concerns appropriately.
>
>
> ## 1. Background knowledge of EMCM method
>
> **The background knowledge of EMCM method in general active learning is discussed in Appendix B.3.** We introduce a high-level intuitive interpretation of the EMCM principle in AL (Eq.(7)) here. **The model change is a reasonable indicator for reducing the generalization error for the following two major reasons. First, The generalization capability can be changed if and only if the current model is changed. As a result, it is useless to query the instance that cannot update the current model in AL. Second, The data points significantly changing the current model is expected to produce a faster convergence speed to the true model,** and this is the underlying motivation behind the EMCM framework.
>
>
>
>
> ## 2. Running time performance of the proposed method and other baselines on the large-scale dataset
>
> In fact, **we compare the running time performance of the proposed method and other baselines in Appendix G.1.2, including one large-scale dataset, Reddit.** Figure 7 shows that our method is both accurate and efficient. **Just like the suggestions by Reviewer bxJi, we will move Figure 7 to the main body for better presentation.**
>
>
>
>
> ## 3. Short conclusion
>
> **We will add more discussion in the conclusion part with the limitations of our method. The limitations and future works are discussed in Appendix H. We will also make sure there are no typos.**
>
>
> ## 4. Extensions of the proposed method
>
> For the extension of the link prediction task of our proposed method, we believe that we cannot directly apply the proposed algorithm because it is challenging to derive the new closed-form solution of the acquisition function on the link-level task since the starting Bayesian interpretation of GNNs only works for the node-level task. We leave the exploration of graph active learning methods for link-level tasks as future work. We also leave the exploration of more advanced batch active learning variants of our proposed method as future work. One possible solution is to incorporate the techniques[1] used in the general batch active learning task for EMCM. Both these directions are very promising extensions of our work.
>
>
> [1] Cai, Wenbin, Muhan Zhang, and Ya Zhang. "Batch mode active learning for regression with expected model change." IEEE transactions on neural networks and learning systems 28.7 (2016): 1668-1681.

---

### Official Review · Reviewer_MppE · 2023-07-03

**Soundness:** 3 good
**Presentation:** 4 excellent
**Contribution:** 3 good
**Rating:** 7
**Confidence:** 4

**Summary:**

This paper proposes a new active learning strategy based on EMCM principle under the task of graph node-level semi-supervised predictions. The most significant contributions of this paper are 1) extending the EMCM principle to GNNs leading to the MAP estimate correlated (interpretable) acquisition function on graphs, 2) proposing a regularized single-level optimization process to solve the bi-level optimization problem. The comprehensive experiments demonstrate the efficiency and effectiveness of the acquisition function.

**Strengths:**

- S1: Great presentations.
- S2: Equipping the AL acquisition function with Bayesian interpretability is a promising motivation.
- S3: The techniques and explanations are largely sound.
- S4: Promising experimental results.

**Weaknesses:**

- W1: The name of the method makes no sense: expecteD mOdel Change maximizaTion On gRaphs (DOCTOR). It fails to imply the AL strategy, making people hard to correlate this name with the method. I prefer a clear name: Graph Expected Model Change Maximization (GEMCM) that indicates both the task and method.
- W2: Typo in Page 2, line 55: "efficacy and efficacy"
- W3: Page 3, line 109: "the nodes are not i.i.d. but linked with edges such that connected nodes tend to have the same label." I can't see the correlation between this challenge and the method. I suggest explaining it with the modeling of graphon.
- W4: Line 218: Can you answer what if the hypothetical label is wrong? It can be the case that we have a high probability with a wrong label and a low change, but a low probability with the true label and a high change. It intuitively make no sense. This is a non-trivial limitation and should be mentioned.
- W5: Can you discuss to what extent, your approximation is valid? I mean can you bound the difference between your approximation and the expectation?
- W6: I expect the authors discussing the connection between the proposed method and strict proper scoring rules instead of only the EEM. I also expect the theoretical comparison between the proposed method and entropy minimization (may be in terms of strictly proper scoring rules).
- W7: I'm confused by the reason why the forward propagation in the equation (2) is a constraint, and why this constraint can be transformed in to the optimization problem in Theorem 3.1. I suggest providing intuitive explanation here.
- W8: Line 234: Can you adding an explanation for the reason why we maintain the smallest eigenvalues (high-frequency) and corresponding eigenvectors instead of the largest ones? I noticed literature believes that the high-frequency information leads to out-of-distribution problem. Is this belief correlated to your approximations here?

**Questions:**

My questions are listed in weaknesses.

**Limitations:**

Yes.

---

> ### Author Rebuttal · Authors · 2023-08-08
>
> $$\textcolor{red}{\text{The detailed rebuttal will be given if needed. Please let us know which point needs further clarification after reading this compact response.}}$$
>
> Thanks for such a brilliant and constructive review!
>
> 1. Model name. The new name is a great suggestion due to the implied interpretable meaning. We will change it to GEMCM.
>
>
> 2. Typo. We will change the second "efficacy" to "efficiency".
>
> 3. Graphon modeling. In fact, the non-i.i.d challenge is indeed not directly relevant to our proposed method. We include this challenge mainly because it hinders the direct application of general active learning techniques. We do agree that graphons can come into the picture as a mathematical tool to model large and complex graphs, capturing the non-i.i.d. characteristics of graph-structured data, including dependencies or homophily. We will include a discussion on the graphons.
>
> 4. Hypothetical label. The hypothetical label is an artificially-assigned temporary label for an unlabeled node or sample when selecting the query node from the unlabeled pool. It is only an introduced term in the look-ahead model for active learning, and it can be assigned as an arbitrary label in the label space. Eq.(7) iterates all possible hypothetical labels $y_k^+$, and gets the corresponding model change $u(\tilde{\Theta}^*) - u(\Theta)^*$, and weights it with the corresponding predictive probability $\mathbb{P}(y_k^+ | k)$. Therefore, whether the hypothetical label is aligned with the actual hidden ground-truth label is not relevant here in the node selection stage, and we only care about how large will the final expected model change be if we add each candidacy node from the unlabeled pool into the next round of training.
>
>
>
> 5. Bound on approximation error.
> For the truncated spectral approximation, let us denote that the original Laplacian matrix $\tilde{\mathbf{L}} \in \mathbb{R}^{n \times n}$ can be decomposed as $\tilde{\mathbf{L}} = \bar{\mathbf{V}}\bar{\boldsymbol{\Lambda}}\bar{\mathbf{V}}^T$. We only keep the top-$m$ smallest eigenvalues in $\bar{\boldsymbol{\Lambda}}$ and denote the new matrix as $\boldsymbol{\Lambda} \in \mathbb{R}^{m \times m}$. Accordingly, the corresponding eigenvectors are kept in $\bar{\mathbf{V}}$ so we get $\mathbf{V}\in\mathbb{R}^{n \times m}$. Then we can now project the node embedding $\mathbf{u}$ onto the space spanned by these truncated $m$ eigenvectors instead of all the $n$ eigenvectors. Therefore, the approximation error can be formulated as $\|\mathbf{V}^T\mathbf{u} - \bar{\mathbf{V}}^T\mathbf{u}\|_2$. Assume the node embedding is bounded $\|\mathbf{u}\|_2 \leq C$. We have,
> $$\|\mathbf{V}^T\mathbf{u} - \bar{\mathbf{V}}^T\mathbf{u}\|_2 = \|(\mathbf{V}^T - \bar{\mathbf{V}}^T)\mathbf{u}\|_2  \leq \|(\mathbf{V}^T - \bar{\mathbf{V}}^T)\|_F \|\mathbf{u}\|_2 \leq C \|(\mathbf{V}^T - \bar{\mathbf{V}}^T)\|_F = C \sqrt{n-m}.$$
>
>
> For Laplacian approximation, we apply it on top of $\mathbb{P}(\boldsymbol{\alpha} \mid \mathbf{y})$. According to the Bernstein-von Mises Theorem, under some regularity conditions and the prior is positive, bounded and twice differentiable, we have $$\text{sup}_{\boldsymbol{z}} |\mathbb{P}(\boldsymbol{\alpha} \leq \mathbf{z} | \mathbf{y}) - \mathbb{P}(\tilde{\boldsymbol{\alpha}} \leq \mathbf{z})| \stackrel{\text{a.s.}}{\approx} 0.$$
>
> Here, $\boldsymbol{\alpha}$ is the true posterior without closed-form exact solution and $\tilde{\boldsymbol{\alpha}}$ is the mean of the approximated Gaussian distribution via Laplacian approximation in Theorem 3.3. This result shows that our approximation is quite good from the aspect of the gap in terms of the cumulative distribution function.
>
>
>
> 6. Strict proper scoring rules.
> Thanks to the generalized active learning acquisition function (Eq.(1) in [1]), the lower bound function $\mathcal{A}' (k)$ of our original acquisition function  $\mathcal{A}(k)$ can now be incorporated into this framework. Other methods can also be incorporated into this generalized active learning acquisition function as well. If we set the functional $Q(\mathbb{P}(\boldsymbol{\alpha} | L)) = I(\mathbb{P}(\boldsymbol{\alpha} | L))$ as the mutual information, we can obtain the acquisition function used in [2] that is based on Shannon's entropy. If we set the functional as Eq.(6) in [1], we can get the acquisition function based on the strict proper scoring rules.
>
> [1] Tan, Wei, Lan Du, and Wray Buntine. "Diversity enhanced active learning with strictly proper scoring rules." Advances in Neural Information Processing Systems 34 (2021): 10906-10918.
>
>
>
> 7. Constraints in Eq.(2).
> In Eq.(2), we first do one forward pass of SGC once, and then try to optimize the resulting training loss on the high level without caring about any further forward passes of SGC. Therefore, this one-time forward pass of the SGC model becomes a constraint in Eq.(2) because when we directly optimize the training loss over $\boldsymbol{\Theta}$, the output $\mathbf{U}(\boldsymbol)$ by the forward pass of SGC must be fixed. Theorem 3.1 focuses on the forward pass of SGC only. Theorem 3.1 provides an optimization perspective of the output (node embeddings) by the forward pass of SGC. Intuitively speaking, when we do forward pass of SGC, we inexplicitly try to minimize the total variations of node embeddings and the node embeddings must be smooth over this graph after the forward pass of SGC model, meaning that the node embeddings for two adjacent nodes should be similar. This corresponds to the homophily assumption of the graphs.
>
>
> 8. Maintain the smallest eigenvalues
> In fact, maintaining the smallest eigenvalues corresponds to applying a low-pass filter. Therefore, when we only keep the smallest eigenvalues, we actually remove high-frequency noises in the graph so that we can keep the most significant information in this graph. Hence, the approximation is completed in this manner. That is why we maintain the smallest eigenvalues.

---

> > ### Comment · Reviewer_MppE · 2023-08-10
> >
> > Thank you for your insightful rebuttal. After your explanations, most of my concerns and confusions have been addressed. I'll update my score to from 5 to 7: Accept. Please include all the modifications, explanations, and discussions in your final version.

---

> > > ### Author Response · Authors · 2023-08-17
> > >
> > > Thanks again for your support for our work! We sincerely appreciate your insightful feedback and adjustment in your score. We will definitely keep improving our paper based on all of your constructive suggestions.

---

### Official Review · Reviewer_zbTy · 2023-07-06

**Soundness:** 3 good
**Presentation:** 3 good
**Contribution:** 3 good
**Rating:** 7
**Confidence:** 5

**Summary:**

The authors present yet another active learning approach for graph neural networks, claiming it to be novel. They attempt to build upon the classic expected model change maximization method but in a general active learning setting. To do so, they introduce a Bayesian interpretation of the SGC model, which supposedly unifies the training process of GNN models. Some approximation techniques are used, along with some theoretical analysis.

**Strengths:**

1. The novelty of this paper is ok. It extends the existing literature on general active learning by being the first work to apply EMCM to graphs. The unified Bayesian view also brings some new insights.
2. The method is well-motivated, with an intriguing theoretical analysis. The proposed method is also efficient both from the theoretical aspect and the experimental aspect.
3. The writing is satisfying, with clear logic and notations. I believe this work can benefit researchers from both the GNN domain and the active learning field.

**Weaknesses:**

1. More discussions on the deep connection between the Bayesian interpretations of GNNs and the proposed active learning method DOCTOR should be provided.
2. More background knowledge on the look-ahead models in active learning literature should be provided.
3. More details on the experiments, like the hyperparameters settings, should be provided.

**Questions:**

1. What are the inner connections between the unified Bayesian interpretations of GNNs and the subsequent proposed active learning method? A more compact explanation is preferred.
2. What are the look-ahead models in Section 3.2.3? The term is frequently used, but little context is provided about this term. Is it the potentially trained model after some labels are given by the oracle?
3. What about some possible extensions of this work? How does the proposed method deal with the noisy oracle?

**Limitations:**

There are no major limitations of this work. However, the reviewer has some open discussions about the extensions. For example, what if the label has noise? In real-world cases, there is no oracle in any application since even human experts can sometimes make mistakes. How can this model be adapted to the noisy oracle case? The reviewer would like to see the authors' thoughts on this question.

---

> ### Author Rebuttal · Authors · 2023-08-08
>
> Thanks for your constructive review! We are glad to know that you found our paper novel, well-motivated, and well-written. We hope these responses will address your concerns appropriately.
>
>
> ## 1. Connections between the unified Bayesian interpretations of GNNs and the subsequent proposed active learning method
>
> The inner connections between the unified Bayesian interpretations of GNNs and the subsequent proposed active learning method are as follows. Under semi-supervised settings, current GNN models typically lack a Bayesian probabilistic interpretation. But in active learning,  Bayesian interpretation lays the groundwork for performance-based methods, where our proposed method belongs. It incorporates prior knowledge and updates the posterior beliefs about the model parameters or predictions, assuming if new labeled data becomes available. Our method bridges this gap. More discussions can be found in Sec. 2.4.1.
>
>
> ## 2.Background knowledge on the look-ahead models
>
> The performance-based methods usually consider the look-ahead model. We first hypothetically assume one candidate node $x_+^k$ from the pool of the unlabeled nodes is chosen for query, and its label is revealed as $y_+^k$ by the oracle. Then we analyze the potential influence of this candidate node ($x_+^k, y_+^k$) on the model's parameters or predictions if it is added to the labeled node set for the next round of training. **In fact, more discussions regarding the background knowledge of look-ahead models can be found in Appendix B.3.1.**
>
>
> ## 3. Possible extensions of this work
>
> Thanks for your insightful suggestion. One possible extension is to incorporate the case of a noisy oracle. For example, **we may consider integrating the techniques used in [1] to handle label noise in the oracle. We will leave it for future work since it is currently out of the scope of this work.**
>
> [1] Zhang, Wentao, et al. "Rim: Reliable influence-based active learning on graphs." Advances in Neural Information Processing Systems 34 (2021): 27978-27990.

---

### Official Review · Reviewer_bxJi · 2023-07-10

**Soundness:** 3 good
**Presentation:** 4 excellent
**Contribution:** 3 good
**Rating:** 7
**Confidence:** 4

**Summary:**

The authors propose an active learning method for GNNs, extending the Expected Model Change Maximization (EMCM) principle to GNNs. A Bayesian interpretation for the node embeddings generated by GNNs under the semi-supervised setting is presented. By establishing a connection with expected prediction error minimization, theoretical guarantees for AL performance are driven. The numerical experiments shows improved performance and runtime compared to the existing approaches.

**Strengths:**

The paper is very well-written and clearly motivated. The overall proposed active learning approach makes sense and seems sound (although I haven't check the proof details). I like the proposed approach for deriving Bayesian interpretation for GNNs. There are sufficient experiments to support the claims as well.

**Weaknesses:**

I don't see any glaring weakness other than Figure 1 not including the IGP (the closest model to the proposed in terms of performance). Probably better to move Figure 7 in appendix to the main manuscript. A minor comment: some of the equations in the appendix are out of bound.

**Questions:**

- It would be nice to add an ablation study about the approximations made in the model. I am curious to know the effect of number of retained eigenvalues/vectors in Laplacian approximation on the performance of the model and it relation to homophily (from a graph signal processing point of view, retaining smallest eigenvalues correspond to low-pass filtering of the signal over graph).

**Limitations:**

See above sections.

---

> ### Author Rebuttal · Authors · 2023-08-08
>
> Thank you for your very thoughtful and constructive review. We appreciate the recognition of our active learning approach and the Bayesian interpretation of GNN training. We are glad to know that you found our paper clear, well-motivated, and supported by sufficient experiments. We would also like to thank you for your suggestions for improvement and have addressed each of your points below. We hope these responses will address your concerns appropriately.
>
>
> ## 1. Figure 1 and Figure 7.
>
> We admire your suggestion. **We will shift Figure 7 from Appendix to the main manuscript and probably consider substituting Figure 7 for Figure 1.** In this way, we can better show the accuracy and running time comparison simultaneously in one figure and highlight the efficacy and efficiency of our method.
>
> ## 2. Out of Bound Equations in the Appendix.
>
> Thank you for bringing this to our attention. **We apologize for this oversight (lines 894,910) and will make sure to reformat the equations**.
>
>
> ## 3. Ablation Study.
>
> Your suggestion for adding an ablation study on analyzing the effect of the number of retained eigenvalues/vectors is insightful. In particular, **we actually did a similar sensitivity analysis on the number of retained eigenvalues $m$ in Appendix G.3,** along with the balancing factor $\lambda$. For a clearer presentation, we summarize the case when fixing $\lambda = 5e-4$ with varying numbers of retained eigenvalues $m$ in the following table. More results can be referred to in Appendix G.3. **A larger $m$ usually leads to better performance since the approximation of the node embeddings projection will be more precise, but the improvement may be quite marginal if $m$ is too large with higher computational cost as well.**
>
> | #Retained eigenvalues $m$  |  10  |  20  |  50  |  100 |  200 |
> |:--------------------------:|:----:|:----:|:----:|:----:|:----:|
> |    Accuracy on Cora (%)    | 82.2 | 82.7 | 84.0 | 84.3 | 84.4 |
> |  Accuracy on Citeseer (%)  | 71.7 | 73.0 | 74.5 | 75.5 | 75.7 |
>
> ## 4. Relation to homophily.
>
> We highly appreciate your suggestions. Indeed, **we choose to retrain the top-$m$ smallest eigenvalues based on this exact motivation that it corresponds to a low-pass filter, and the graph signal is assumed to be smooth (a.k.a, it is a homophilous graph).** Therefore, when the homophily principle holds and the graph signals are smooth, **applying a low-pass filter by retaining the smallest eigenvalues in the truncated spectral projection could capture most of the important information present in the graph signal while reducing the computational cost significantly.** This makes it a relevant theoretical understanding of GNN models[1][2]. We will discuss more on its relationship with the homophily concept from graph signal processing (GSP) in Section 3.2.2, and illustrate why we choose to retain the smallest eigenvalues during Laplacian approximation more clearly. **For a formal theoretical analysis of the direct relationship between the model performance and the number of retained smallest eigenvalues from the GSP aspect, we leave it for future work since we need many more assumptions on the graph itself**, like how the graph is generated and the how the label is generated from the graph. Nonetheless, the investigation of this direction is very promising and can motivate new design of the active learning methods for GNNs.
>
>
> [1] Ma, Yao, et al. "A unified view on graph neural networks as graph signal denoising." Proceedings of the 30th ACM International Conference on Information & Knowledge Management. 2021
>
> [2] Dong, Xiaowen, et al. "Graph signal processing for machine learning: A review and new perspectives." IEEE Signal processing magazine 37.6 (2020): 117-127.

---

> > ### Comment · Reviewer_bxJi · 2023-08-17
> >
> > Thanks authors for their response addressing my questions. Having read all reviews and responses, I'm keeping my score as is.

---

> > > ### Author Response · Authors · 2023-08-18
> > >
> > > Thanks again for your strong support for our work! We will improve our paper with all of your insightful suggestions.

---

### Decision · Program_Chairs · 2023-09-21

**Decision:**

Accept (spotlight)

**Comment:**

All reviewers acknowledged the contributions of this paper, so I would recommend to accept as spotlight.